# Gulf of Alaska ice-marginal lake area change over the Landsat record and potential physical controls

Hannah R. Field[1,2], William H. Armstrong[1], and Matthias Huss[3,4,5]

[1]Department of Geological and Environmental Sciences, Appalachian State University, Boone, NC 28607, USA

[2]School of Earth Sciences, The Ohio State University, Columbus, OH 43210, USA

[3]Laboratory of Hydraulics, Hydrology and Glaciology (VAW), ETH Zurich, Zurich, 8093, Switzerland

[4]Swiss Federal Institute for Forest, Snow and Landscape Research (WSL), Birmensdorf, Switzerland

[5]Department of Geosciences, University of Fribourg, Fribourg, Switzerland

Correspondence to: armstrongwh@appstate.edu

**Abstract.** Lakes in contact with glacier margins can impact glacier evolution as well as the downstream biophysical systems, flood hazard, and water resources. Recent work suggests positive feedbacks between glacier wastage and ice-marginal lake evolution, although precise physical controls are not well understood. Here, we quantify ice-marginal lake area change in understudied northwestern North America from 1984 - 2018 and investigate climatic, topographic, and glaciological influences on lake area change. We delineate timeseries of sampled lake perimeters (n = 107 lakes) and find that regional lake area has increased 58 % in aggregate, with individual proglacial lakes growing by 1.28 km$^2$ (125 %) and ice-dammed lakes shrinking by 0.04 km$^2$ (- 15 %) on average. A statistical investigation of climate reanalysis data suggests that changes in summer temperature and winter precipitation exert minimal direct influence on lake area change. Utilizing existing datasets of observed and modelled glacial characteristics, we find that large, wide glaciers with thick lake-adjacent ice are associated with the fastest rate of lake area change, particularly where they are undergoing rapid mass loss in recent times. We observe a dichotomy in which large, low-elevation coastal proglacial lakes have changed most in absolute terms, while small, interior lakes at high elevation changed most in relative terms. Generally, the fastest changing lakes have not experienced the most dramatic temperature or precipitation change, nor are they associated with the highest rates of glacier mass loss. Our work suggests that, while climatic and glaciological factors must play some role in determining lake area change, the influence of a lake's specific geometry and topographic setting overrides these external controls.

**1 Introduction**

The development and evolution of ice-marginal lakes (both proglacial and ice-dammed lakes) may have implications for both upstream glacier systems and downstream fluvial environments (Baker et al., 2016; Otto, 2019; Tweed & Carrivick, 2015). The formation and growth of a proglacial lake (a lake that forms downstream of a glacier terminus) marks a fundamental transition in alpine landscapes, with the intervening lake modifying transport of water, sediment and nutrients to the downstream river, and altering mass loss and dynamics of the upstream glacier (Baker et al., 2016;

Bogen et al., 2015; Dorava & Milner, 2000; Jacquet et al., 2017; Ratajczak et al., 2018). Additionally, the presence of ice-dammed lakes (lakes dammed by a glacier that often form in tributary valleys or at the glacier margin) enables glacial outburst floods (GLOFs) that contribute to short-term changes in downstream geomorphologic and hydrologic dynamics and may pose a serious hazard to downstream communities (Carrivick & Tweed, 2016; Roberts et al., 2003; Tweed & Russell, 1999). The response of ice-marginal lakes, both in terms of number and size, to climate change is

an important issue for alpine environments globally because of these inter-system links (Stokes et al., 2007; Zemp et al., 2015). Despite the critical role of these lakes, little is known about physical controls on ice-marginal lake formation and evolution (Falatkova et al., 2019; Magnin et al., 2020). To address this knowledge gap, we investigate trends in ice-marginal lake area change across northwestern North America, a relatively unstudied region, over the satellite record and explore physical controls on observed behavior.

Globally, proglacial lakes have expanded and increased in number over the 20th-21st centuries (Shugar et al., 2020; Stokes et al., 2007; Tweed & Carrivick, 2015; Wang et al., 2015). Iceland has experienced an increase in number of proglacial lakes, with individual lakes increasing in area by up to 18 km$^2$ (Canas et al., 2015; Tweed & Carrivick, 2015). Across the Hindu Kush Himalaya, glacial lake change has been variable and appears to be indirectly linked to glacier change (Gardelle et al., 2011). Glacial lakes in the Central and Eastern Himalayas have significantly expanded

both in number and size over the past 30 – 40 years, which coincides with glacier retreat and precipitation changes in those regions (Bajracharya et al., 2015; Gardelle et al., 2011; Khadka et al., 2018; Shukla et al., 2018; Treichler et al., 2019; Wang et al., 2015; Zhang et al., 2019). In the Western Himalayas where glaciers are experiencing less retreat, lakes appear to be shrinking (Gardelle et al., 2011). In the southern Andes, glacier lakes (including some lakes not in direct contact with glaciers) appear to be primarily growing in number, with smaller cumulative area increase (7%)

than seen elsewhere (Wilson et al., 2018). Less is known about ice-marginal lakes in northwestern North America, a region that is experiencing increasing air temperatures and changing precipitation that has generally resulted in negative glacier mass balance (Larsen et al., 2015) and loss of glacier coverage (Arendt et al., 2009). Wolfe et al. (2014) indicate that Alaska glacier-dammed lakes have become less common over 1971-2008. The total number of ice-dammed lakes decreased by 23 %, though 34 % of lakes existing in 2008 were newly formed (Wolfe et al., 2014).

We expand upon the work of Wolfe et al. (2014) by assessing change on proglacial lakes in addition to ice-dammed lakes, characterize area change in addition to quantity, and probe the underlying physical controls.

The development and evolution of an ice-marginal lake can impact its associated glacier. Theoretically, the presence of proglacial lakes can influence glacier ablation through thermal and mechanical processes (Tweed & Carrivick,

2015). Observations of the glaciological impact of lake formation is mixed, with some studies finding increased rates of mass loss (King et al., 2019; 2020) and speed (Tsutaki et al., 2011; Watson et al., 2020) on lake-terminating glaciers, with support from non-numerical modeling (Sutherland et al., 2020), though other observational studies suggest glacier-averaged mass balance is minimally affected by the presence of a proglacial lake (Larsen et al., 2015). The presence of a lake at the terminus of a glacier may allow thermally-induced subaqueous melt (e.g., Robinson and Matthaei, 2007) and may also increase glacier mass loss by enabling increased calving (e.g., Chernos et al., 2016) and/or by modulating subglacial hydraulics (Tsutaki et al., 2011). However, despite their similarity to marine-terminating (tidewater) glaciers, lake-terminating glaciers likely calve less vigorously and experience less subaqueous melt than their tidewater counterparts due to shallower and colder water near the terminus and the lack of upwelling meltwater plumes (Truffer & Motyka, 2016).

Above, we discuss the impacts of ice-marginal lake change on their associated glacier, but this is a two-way process, with glacier change also impacting their associated lakes. Globally, the extensive retreat of glaciers been associated with the increase of the number and size of proglacial lakes (Otto, 2019; Stokes et al., 2007). However, the exact mechanisms driving lake area change and its sensitivity to climate change are not well understood. Glacier processes (e.g., sensitivity of glacier mass balance to temperature change, glacier response time) and local subglacial topography both likely contribute to how lakes change over time and space (Debnath et al., 2018; Otto, 2019; Song et al., 2017), and these factors themselves may interact and/or change over time. Previous work suggests that the main factor in lake development is the presence of glacial overdeepenings and confining topography (Buckel et al., 2018; S. Cook & Swift, 2012; Farías-Barahona et al., 2020; Haeberli et al., 2016; Otto, 2019). Changing air temperature and precipitation also play an important role in proglacial lake area change by influencing glacial thinning, retreat, and meltwater runoff (Debnath et al., 2018; Treichler, et al., 2019), though Brun et al. (2020), focusing on closed basins not in direct contact with glaciers minimal influence of glacier mass loss on Tibetan lake volume change. In Alaska, glacier thinning and tributary disconnection alter basin morphology, and the distribution of ice-dammed lakes shifted to higher elevation over the late 20[th] century (Wolfe et al., 2014). Glaciological factors such as debris cover and regional glacier mass loss may also influence proglacial lake evolution (Song et al., 2017).

Ice-marginal lakes can impact downstream ecosystems by altering sediment fluxes, geochemical cycling, and downstream geomorphological characteristics, among other impacts (Baker et al., 2016; Dorava & Milner, 2000). The reduced suspended sediment load in glacier-fed streams and rivers downstream from proglacial lakes enhances habitats for aquatic organisms (Bogen et al., 2015; Dorava & Milner, 2000). Stream temperature is higher and less time variable downstream from lakes, and this thermal regulation is also beneficial for many aquatic species (Dorava & Milner, 2000; Fellman et al., 2014). Proglacial lakes may also stabilize downstream channel morphology, contributing increased bank stability (Dorava & Milner, 2000). Conversely, ice-dammed lakes may increase the rate of channel migration and contribute to more transient channel geometry due to outburst flooding (Jacquet et al., 2017). Understanding the development and evolution of these lakes is critical due to their influence both local and regional environments.

The complicated interrelations of geomorphic, climatic, and glaciologic influences on ice-marginal lake area change must be untangled to develop a better understanding of the main drivers of ice-marginal lake area dynamics. A conceptual model for physical controls on both proglacial and ice-dammed lake behavior is necessary for predicting their evolution in a warming world, highlighting which lakes may be most sensitive to perturbations, and assessing potential impacts on their adjacent biophysical systems. This study seeks to narrow this knowledge gap in two ways. First, we document what is happening – how are proglacial lakes changing across northwestern North America? What are the rates and spatial patterns of change? Secondly, we investigate why this is happening – what are the dominant physical controls on ice-marginal lake behavior? Do these controls vary across space? Explicitly, we employ statistical analyses to explore climatic, glaciological, and topographic associations with ice-marginal lake area change. By answering the questions above, we hope to inform our understanding of this critical landscape interface to enhance prediction of how upstream and downstream systems will evolve in a warming world.

## 2 Study area, datasets, and data pre-processing

Below, we introduce the study region and then describe our climatic, glaciologic, and geomorphic data sources for statistical analyses employed to investigate drivers of lake area change. In Secs. 2.2-2.4, we describe the datasets used to evaluate potential control variables for ice-marginal lake area change (Table 1). Later in the manuscript, we use the terms "environmental variables" or "predictor variables" to collectively describe these climatic, glaciologic, and topographic descriptors of each lake's setting.

### 2.1 Study area

We study a sampling of ice-marginal lakes (n = 107) that span 48 – 68 °N and 116 – 154 °W, covering much of northwest North America along the Gulf of Alaska and into the interior. The lakes are found from the Brooks Range, to the Washington Cascades, and Canadian Rockies, and are located in the states and provinces of Alaska, Washington, Yukon Territory, British Columbia, and Alberta (Fig. 1). The region is extensively glacierized (101,700 km$^2$) and contains 14% of the world's glaciers and ice caps (GIC) by area (Randolph Glacier Inventory Regions 01 and 02; Gardner et al., 2013). Glaciers across northwestern North America are losing mass faster than any other region (-73 Gt a$^{-1}$ or -0.85 m w.e. a$^{-1}$ for Alaska; -12 Gt a$^{-1}$ or -0.83 m w.e. a$^{-1}$ for Western Canada and continental USA; Zemp et al., 2019) and account for 26% of GIC contributions to sea level rise, despite comprising only 14% of global GIC volume (Zemp et al., 2019). Despite this general picture of glacier wastage, significant spatial and temporal variability exists in the pattern of glacier mass loss (Menounos et al., 2019).

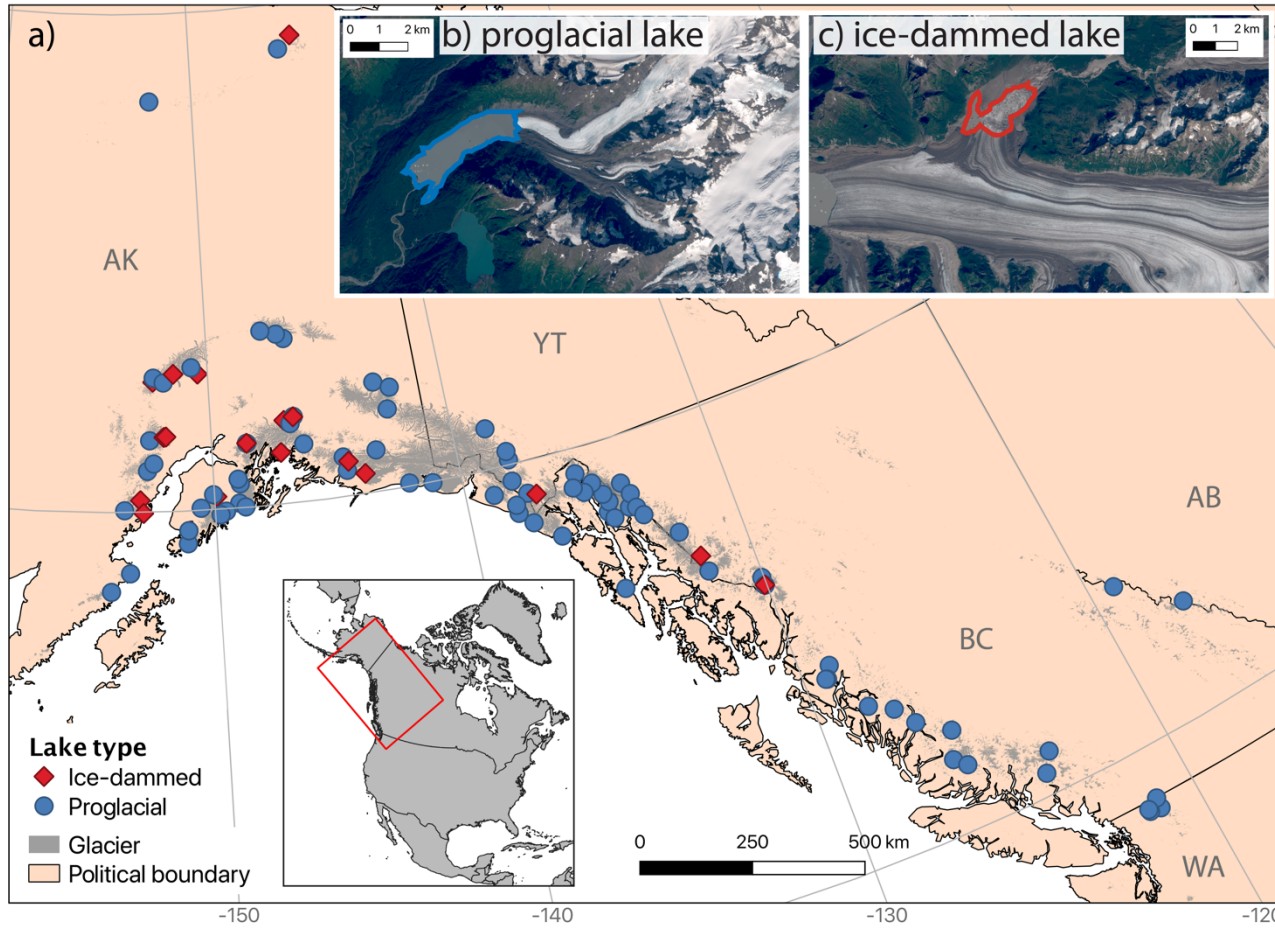

**Figure 1. a) Map of study region showing sampled proglacial (blue circles) and ice-dammed (red diamonds) lakes. Glacier extent is shown as gray fill (RGI, 2017) and black lines indicate political boundaries. Two-letter state and province abbreviations are given in gray text. The inset map shows the location of the study area (red box) in northwestern North America. Examples are shown of b) a proglacial lake (Unnamed lake downstream from Twentymile Glacier; 60.94 N, -148.78 E) and c) an ice-dammed lake (Van Cleve Lake dammed by Miles Glacier; 60.70 N, -144.41 E). Background imagery in b) and c) is from Landsat 8.**

## 2.2 Climatic variables

We retrieve climate data from the Scenarios Network for Alaska + Arctic Planning (SNAP) database (accessible at http://ckan.snap.uaf.edu/dataset). The database includes 2 km × 2 km resolution gridded climate data downscaled from the Climate Research Unit Time-series (CRU TS3.10) and Parameter-elevation Regressions on Independent Slopes Model (PRISM) datasets (Daly et al., 1997; Harris et al., 2014). SNAP provides access to historical air temperature estimations including seasonal, annual, and decadal monthly means. We retrieve decadal summer air temperature and winter precipitation data (Table 1; Fig. S1a-b), which are the most relevant quantities to non-equatorial glacier mass balance (Cuffey & Paterson, 2010). The summer temperature products have average uncertainties of +/- 0.3ºC, with a 0.1º C cold bias (Bieniek et al., 2016). Precipitation data includes estimates of monthly totals and means of annual,

seasonal, and decadal monthly means of total precipitation. The winter precipitation estimates have an uncertainty of +/- 4.1 mm d$^{-1}$, with a -0.9 mm d$^{-1}$ dry bias (Bieniek et al., 2016). These data do not subdivide precipitation into rain and snow. We investigated the influence of 10-year averages of winter (December, January, and February) precipitation, summer (June, July, and August) air temperature, and the changes in these quantities between the 2000-

2009 decade and the 1960-1969 decade (Fig. S1c-d). We utilize the 1960s decade to consider the longest-term comparison allowed by the SNAP dataset. We manually measure the shortest distance between each lake and a simplified representation of the Gulf of Alaska coastline (Fig. S2) to provide a metric for a lake's continentality.

**Table 1. Climatic, glaciologic, and topographic datasets and respective variables retrieved and used in our analyses.**

| Source | Variables Retrieved |
|---|---|
| **Scenarios Network for Alaska + Arctic Planning (SNAP)** | Summer air temperature (1960's, 1980's, 2000's - Jun, Jul, Aug decadal average) <br> Winter precipitation (1960's, 1980's, 2000's - Dec, Jan, Feb decadal average) |
| **USGS GTOPO30** | Elevation |
| **Randolph Glacier Inventory (RGI v6.0)** | Glacier geometry (glacier area, minimum, maximum, and median elevation of glacier, mean slope of glacier surface, orientation of glacier surface, length of longest flowline on glacier) |
| **Farinotti et al. (2019) ice thickness product** | Mean, maximum, and standard deviation of ice thickness across glacier, glacier volume, near terminal ice thickness |
| **Huss and Hock (2015) mass balance dataset** | Mean annual mass balance (1980's, 1990's, 2000's, 2010's), summed balance 1980's - 2010's, terminal balance, winter accumulation, glacier response time, mass balance gradient |


## 2.3 Glaciologic variables

Glaciologic variables may be subdivided into variables that describe glacier geometry and those that describe glacier mass balance. To investigate the influence of geometric attributes of each lake's adjacent glacier, we use the Randolph Glacier Inventory (RGI) version 6.0, a globally complete, frozen-in-time snapshot of glacier outlines produced to

provide an inventory of glaciers at the start of the twenty-first century (Pfeffer et al., 2014; RGI 2017). The RGI also provides glacier geometrical characteristics, including glacier area, elevation, mean surface slope, flow direction, and the length of the longest flowline. Additionally, we use information on glacier ice thickness based on the Farinotti et al. (2019) consensus ice thickness product. This dataset relies on glacier surface characteristics of RGI glaciers to produce predicted ice thickness distributions from an ensemble of up to five models (Farinotti et al., 2019). The

ensemble approach produces ice thickness estimates that are more robust and accurate than any individual model, with 50 % of all modeled mean ice thickness agreeing with observations to within +30/-20 % (Farinotti et al., 2019). Despite this overall agreement, local deviations up to two times the observed ice thickness do exist (Farinotti et al., 2019). We further process these data to compute metrics such as the mean, median, maximum ice thickness of each glacier, as well as its total volume.


To assess the influence of glacier mass balance on ice-marginal lake area change, we use data from Huss and Hock (2015), who estimated mass balance distribution for individual RGI 6.0 glaciers for the period 1980 – 2016 based on the Global Glacier Evolution Model (GloGEM). GloGEM employs a calibrated temperature-index model driven by ERA-interim re-analysis climate data. Huss and Hock (2015) report that 66 % of modeled net annual mass balance

estimates agree with observations to within +/- 0.25 m w.e. a$^{-1}$. For the estimates that fall outside of this range, smaller glaciers are more prone to mass balance overestimates than large glaciers (Huss & Hock, 2015). From this dataset, we investigate variables that characterize annual mass balance, cumulative mass balance, near terminal mass balance, glacier response time, and mass balance gradient. Glacier response (T) time has been determined based on the strongly simplified context proposed by Johannesson et al. (1989) based on maximum ice thickness and mass balance at the

glacier terminus as $T = -H_{max}/b_t$ where $H_{max}$ is the maximum thickness of the glacier and $b_t$ is mass balance of the lowermost elevation band (10 m) of the glacier (Jóhannesson et al., 1989; Huss and Hock, 2015). Mass balance gradients have been determined by a linear fit with elevation through computed mass balances in the ablation area for each year individually as an average over the entire study period.

**2.4 Topographic variables**

We extract surface elevation data from the U.S. Geological Survey (USGS) GTOPO30, a 1 km resolution global digital elevation model (Danielson & Gesch, 2011). For the United States, GTOPO30 utilizes the USGS digital elevation models and in Canada utilizes the Digital Terrain Elevation Data and the Digital Chart of the World datasets. The relatively coarse resolution of this dataset is sufficient for the purpose of providing a general estimate of lake

surface elevation.

For each lake-associated glacier, we extract glacier width as well as the width of its confining valley in the terminus region from Google Earth imagery. The valley width is estimated from ridge to ridge measurement, which we manually identify using an elevation overlay. Near-terminal glacier width is measured at the terminus of the glacier

in contact with the proglacial lake. For ice-dammed lakes, valley width is estimated as the ridge to ridge distance transverse to the dammed valley axis. Near terminal glacier width in ice-dammed settings is approximated as the straight-line length of the glacier-lake interface.

In the previous section we described glacier-wide attributes that may be associated with lake area change. However,

it is plausible that glacier attributes in the region immediately bordering an ice-marginal lake may be more important for the lake's evolution. To assess the influence of local ice thickness, we extract these data for the lake-adjacent region of the glacier associated with each sample lake (Fig. 2). Ice thickness in the lake-adjacent area better reflects the extent to which a subglacial overdeepening exists that can allow for further lake growth. We delineate these lake-adjacent regions using the RGI 6.0 outline and recent satellite data. We then extract the Farinotti et al. (2019) ice

thickness in this zone and compute its statistics. For glaciers associated with proglacial lakes, we define the "near-terminal zone" as the terminal 20 % of the upstream glacier. For ice-dammed lakes, we define the lake-adjacent region as 10 % of the glacier length up- and down-glacier from the lake – glacier junction. We used a fixed relative area (scaled by glacier area) to ensure uniformity across study sites in our definition of the near-terminus zone.

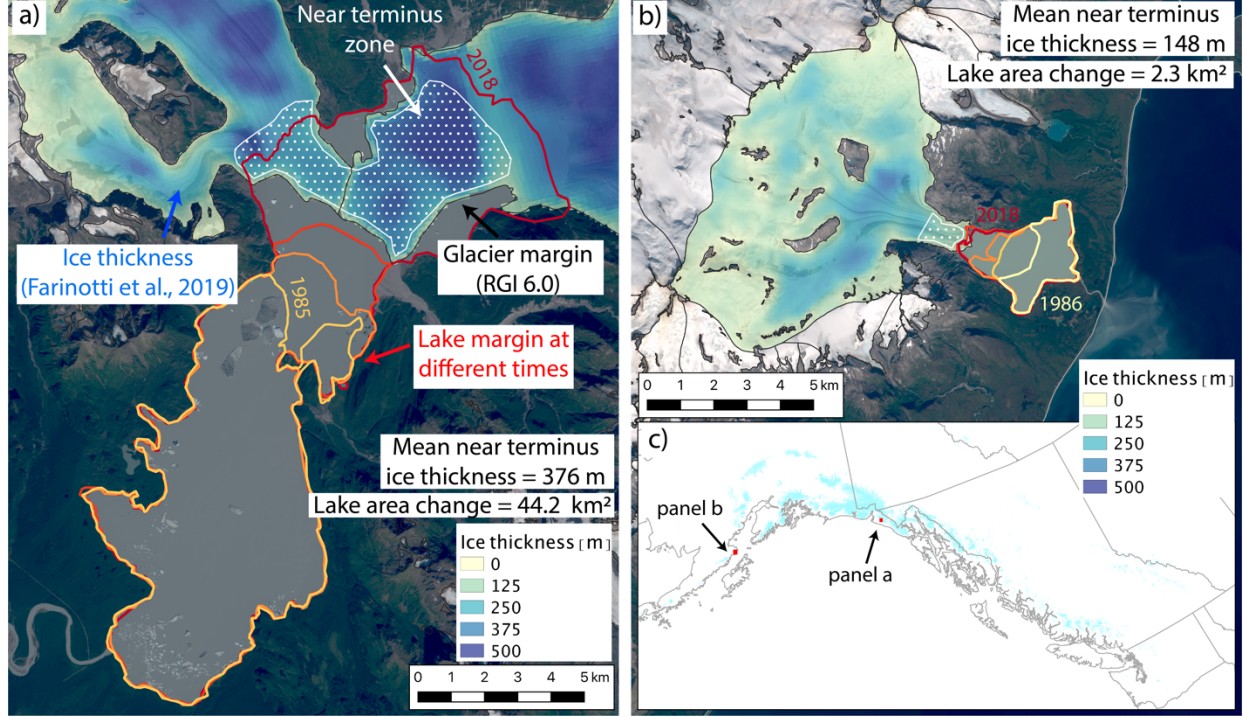

**Figure 2: Illustration of the potential importance of the near terminus topography. (a) Time-varying lake margins (red) and estimated ice thickness distribution (blue) at Harlequin Lake below Yakutat Glacier, Alaska (59.48 N, -138.90 E). Zone for calculating near terminus ice thickness is shown as stippled white, and the RGI 6.0 glacier margin is shown as a black line. This specific glacier-lake system is discussed in Trüssel et al. (2015). (b) Same as in (a), but for an unnamed lake below Fourpeaked Glacier, Alaska (58.77 N, -153.45 E). (c) Overview map showing locations of panels a and b. Ice thickness data are from Farinotti et al. (2019). Glacier outlines are from RGI (2017). Background imagery is from Landsat 8.**

## 3 Methods

Below we describe the procedure we use for sampling and delineating lakes and follow with a description of the analyses we perform to investigate physical controls on the evolution of ice-marginal lake area for our sample set.

### 3.1 Lake sampling and delineation

We use the term "ice-marginal lake" to describe any lake that is in direct physical contact with one or more glaciers, regardless of whether it occurs at a terminal or lateral margin, and independent of dam type (e.g., bedrock, moraine, glacier ice). We use "proglacial lake" to describe an ice-marginal lake that is immediately downstream from a glacier's terminus. We consider an "ice-dammed lake" to be an ice-marginal lake that is found at a glacier's lateral margin and appears to be impounded by glacier ice. Most of the study lakes remained in contact with a glacier for the entire study period, and we discard lakes that detached from their associated glacier from later statistical analyses (described below). Due to the time-consuming nature of high-accuracy manual lake digitization, we do not attempt to delineate every single ice-marginal lake in the study area, but rather sample an evenly distributed subset of lakes to provide an estimate of regional lake area change behavior. We utilize a gridded map and select a similar number of lakes in each grid cell to avoid biased site selection and clustering. A subset of lakes (n = 40) is sampled from a historical catalog

of ice-marginal lakes in Alaska (Post & Mayo, 1971) to avoid undersampling lakes that disappeared and could not be observed in recent satellite imagery. Our dataset for area change analysis includes 107 ice-marginal lakes (88 proglacial and 19 ice-dammed). For statistical analyses, this number is decreased to 73 proglacial lakes and 14 ice-dammed lakes (87 ice-marginal lakes in total) due to the discarded lakes that detached from their associated glacier during the study period. Of the 40 lakes sampled from Post and Mayo (1971), 19 lakes were ice-dammed and the rest of our sample set are proglacial lakes of uncategorized dam material (e.g., moraine, bedrock, or landslide). Our study lakes are generally relatively small, with a median (mean) initial area of 0.78 km$^2$ (4.06 km$^2$). Excluding lakes that appeared during the study period, the median (mean) initial area is 1.08 km$^2$ (4.42 km$^2$), with an interquartile range of 0.26 to 3.66 km$^2$.

We accessed the Landsat 5 – 8 record using Google Earth Engine to estimate lake area change between 1984 – 2018 by utilizing the Google Earth Digitization Tool (GEEDiT) (Lea, 2018). GEEDiT was initially developed by Lea (2018) for delineating glacier termini, however we adapted it to manually digitize lake boundaries from pan-sharpened true color optical imagery (Fig. 1b-c). Adapting GEEDiT for this purpose required a post-processing step to close polylines into polygons, which was accomplished using the Shapely package in Python (Gillies et al., 2007). Each lake's margin was manually digitized between 4 -7 times with intervals of approximately 5 – 10 years separating images (Table S1) for a total of 540 digitized lake outlines. We exclusively utilize summer imagery (June, July, August) to increase confidence in lake perimeter digitization and to minimize the influence of seasonal cycles on our estimates of lake area change.

## 3.2 Lake area change analysis

We determine absolute lake area change ($\Delta A$) as the simple difference in area between our last and first lake delineations, where a positive area change indicates a growing lake. This simple difference means that our characterization of area change is sensitive to the exact value of lake area at the time of image acquisition. This sensitivity will not produce significant error if interannual and seasonal variations in lake area change are small relative to the long-term trend. However, where short-term variability is large relative to the long-term trend, this single-pair area change metric may be less accurate in estimating the true long-term lake area change. This may make our estimates of ice-dammed lake area change more uncertain because these lakes are susceptible to period outburst flooding. We determine relative lake area change as $\Delta A / A_0$, where $A_0$ is the lake's first observed area and $\Delta A$ represents the absolute change in lake area over the study period.

Lake area change takes one of two forms: 1) progression along a continuum, such as a small lake growing larger, or; 2) a system switch, such as the appearance of a new lake, or disconnection of an ice-marginal lake from its associated glacier. We characterize these styles of lake area change in two distinct ways, as described below.

For lakes moving along a continuum, we observe that there are several different patterns of lake area change over time. We quantify these behaviors by categorizing the area change time series of each lake as linear, exponential, or

logarithmic change over the study period. The accuracy of this characterization again assumes that interannual and seasonal variations in lake area are small relative to the long-term trend. This assumption may be problematic for ice-dammed lakes that experience regular outburst flooding resulting in lake drainage followed by a refilling period. Anecdotally, we did not observe any lakes to disappear and then re-appear in our study sample, and so assume this source of error is small in our overall analysis. Further, our main conclusions do not rely heavily on this metric, and

we present it here merely as a tool to explore varied lake change behavior. In addition to the temporal styles of change described above, we defined stable lakes as those with area change of $\leq 0.10$ km$^2$. We use this relatively high stability threshold to produce conservative results that do not classify area change styles unless the signal is large. We interpret linear area change trends to represent steady growth or shrinkage, while exponential trends indicate either accelerating growth or decelerating shrinkage, and logarithmic change suggests decelerating growth or accelerating shrinkage over

time. We utilize the Ezyfit Toolbox in *MATLAB version R2019b* in order to determine the best fitting line type for each lake area change timeseries. Lakes were categorized as having the growth style with the line fit that explains the highest variance in the data (i.e., highest r$^2$, value).

The system switches of new lake appearance or lake disconnection represent the first and final stages of ice-marginal

lake evolution (Emmer et al., 2020). We record the date of the first image in which the lake either appeared or became detached. We denote lakes that appeared during the study period as "new lakes" in later figures but include them with extant lakes for all analyses. We exclude lakes that detached from their adjacent glacier from our lakes area change analyses and investigation of physical controls because they complicate interpretation, particularly where the lake detached early in the study period. However, we retain these lakes in this inventory to represent the late stages of

proglacial lakes in deglaciating environments and the date of their disconnection may yield meaningful insight. Additionally, we observed that some lakes appeared during the study period. We include these lakes in area change analyses and investigations of physical controls because they represent the early proglacial lake growth, and all appeared early in our study period.

**3.3 Correlation testing**

We utilize the non-parametric Kendall correlation test to assess the strength and significance of relationships between lake area change (both absolute and relative) and potential physical control variables. We also employ Kendall correlation to test for statistical relationships between environmental variables. The Kendall test makes no assumption of data normality and is calculated from the rank of data points rather than their actual values, which makes it robust

to outliers (Helsel & Hirsch, 1992). Further, the Kendal test does not assume variables are associated linearly, and can be applied to any monotonic relationship. All of these attributes make the non-parametric Kendall test preferable to parametric tests such as Pearson's linear correlation test because many of our datasets are non-normally distributed, contain outliers, and exhibit non-linear relationships. We also employ the non-parametric Kendall-Theil robust line (a.k.a. Sen slope) to estimate best fit lines that are insensitive to outliers (Helsel & Hirsch, 1992). The Kendall-Theil

robust line is implemented in *MATLAB* through a third party code, available at https://www.mathworks.com/matlabcentral/fileexchange/34308-theil-sen-estimator. We restrict our statistical

analyses to the ice-marginal lakes that remained in contact with their associated glacier(s) throughout the study period (n = 87). We implement an alpha level of 0.1 for testing correlation significance. Analyses are performed using *MATLAB version R2019b* and we use the `corr` function to determine both the significance level (p-value) and Kendall $\tau$ test statistic. Further, we employed principal components (PC) analysis to reduce data dimensionality and test for correlations between lake area change and PC axis scores.

### 3.4 Principal components analysis procedure and interpretation

In addition to the single-variable correlation tests described in Sec. 3.3, we undertake principal components analysis (PCA) to reduce the dimensionality of the datasets of topographic, climatic, and glaciologic factors, many of which are themselves correlated (see Sec. 4.4 for discussion of covariance). To prevent high-valued environmental variables (e.g., glacier area, which can exceed 1000 km$^2$) from dominating dataset variance relative to low-valued variables (e.g., mass balance gradient, which is generally 0.1 – 1 m w.e. per 100 m), we standardize input variables (Table 3) by their minima and maxima to ensure that all variables range from 0 to 1. We then run PCA on the standardized environmental variables using Matlab's `pca` function. We investigate the variance explained by each principal component axis (i.e., "scree plot") and the input variable loadings onto each axis (Table S2). After determining which principal component axes are most relevant (described below), we correlate the lakes' principal component scores to lake area change, using the same procedure described in Sec. 3.3.

### 4 Results

In this section, we first provide summary statistics of lake area change for the subset of northwestern North America ice-marginal lakes considered in this study, both in terms of absolute and relative change. We follow by presenting statistical associations between lake area change and predictor variables such as climate, glacier mass balance, and surrounding topography. Absolute and relative area change have substantially different statistical associations with predictor variables, and we thus discuss these findings in separate sections. We first perform bivariate correlations between individual environmental variables and lake area change, then discuss covariance between individual environmental variables, and finally present multivariate statistical analyses. We present both bivariate and multivariate results because each type of analysis provides context for the other. Comprehensive investigation of both types of statistical tests provides the strongest foundation for interpretation of potential physical controls on ice-marginal lake area change.

### 4.1 Summary of regional lake area change

Of the 107 ice-marginal lakes (both proglacial and ice-dammed) investigated in this study, which does not include every lake in the region, we find that 70 % grew in area, 12 % shrank, and 18 % remained relatively constant, changing by less than ± 0.1 km$^2$ (Figs. 3-4; Table 2). Of proglacial lakes (n = 87), 83 % grew, 5 % shrank, and 13 % remained relatively steady. In contrast, of the 20 ice-dammed lakes, only 15 % increased in area, while 45 % shrank, and 40 % were relatively unchanged (Figs. 3-5; Table 2). Analyzing all ice-marginal lakes together, lake coverage increased in cumulative area by 59 % relative to 1984 (432 to 687 km$^2$). Dividing the study lakes into their sub-classes, proglacial

lakes grew in total area by 81 % (336 to 606 km$^2$) while ice-dammed lake area shrunk by –17 % (96 to 80 km$^2$; Table

355     2).

Individual proglacial lakes experienced a median area change of +1.3 km$^2$ (mean = +3.1 km$^2$), with an interdecile range (10$^{th}$ to 90$^{th}$ percentile) growing between 0.0 and 6.8 km$^2$ (Figs. 4a; Table 2). At the extremes, we observe a minimum proglacial lake area change of –2.4 km$^2$ and maximum of +44.2 km$^2$. In terms of area change relative to

each lake's initial area, we find a median proglacial lake growth of +125 %, with an interquartile range of +42 to +384 % (Figs. 3-4b). Considering the full range of relative area change produces physically meaningless values where lakes did not exist or were very small at the start of the record.

In contrast, ice-dammed lakes in this study experienced a median area change of –0.04 km$^2$ with an interdecile range

of area change from -3.71 to 0.36 km$^2$ (Figs. 3-4a; Table 2). At the extremes, one ice-dammed lake shrunk by –10.8 km$^2$ and one grew by +5.4 km$^2$. In terms of area change relative to each lake's initial area, we find a median ice dammed lake area decline of –15 %, with an interquartile range of –56 to +8 % (Figs. 3-4b).

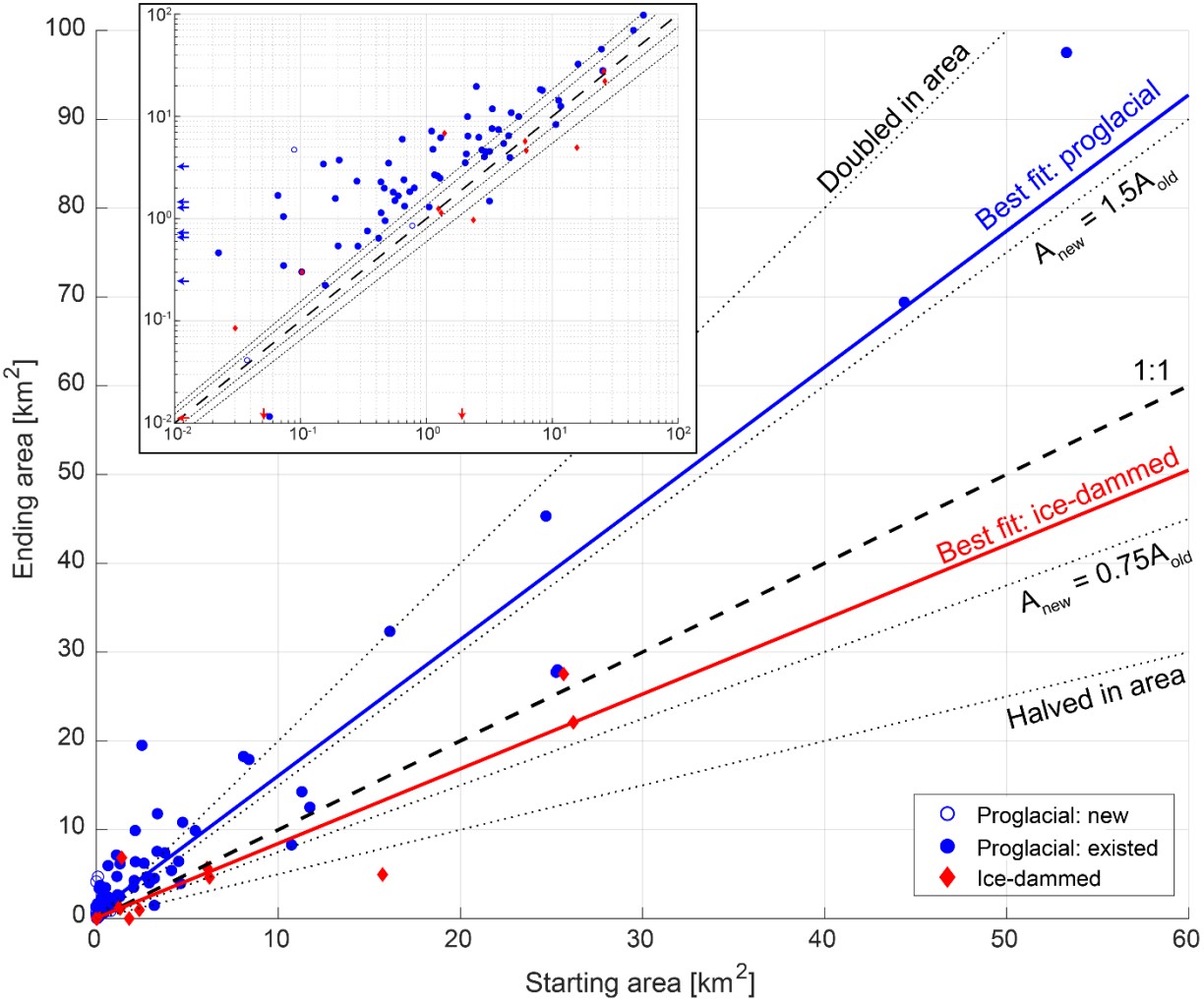

**Figure 3.  Ice-marginal lake area at the start (horizontal axis) and end (vertical axis) of the study period. Proglacial lakes that existed for the entire study period are shown as filled blue circles, while proglacial lakes that appeared that time ("new lakes") are shown as unfilled symbols. Red diamonds depict ice-dammed lakes. The dashed line shows 1:1 (i.e., lakes with constant area), while the dotted lines show various levels of relative area change. The blue (red) solid lines show the Theil-Sen estimator line of best fit to proglacial (ice-dammed) lakes. The inset shows the same data in log-log space to better**

**display the behavior of small lakes.**

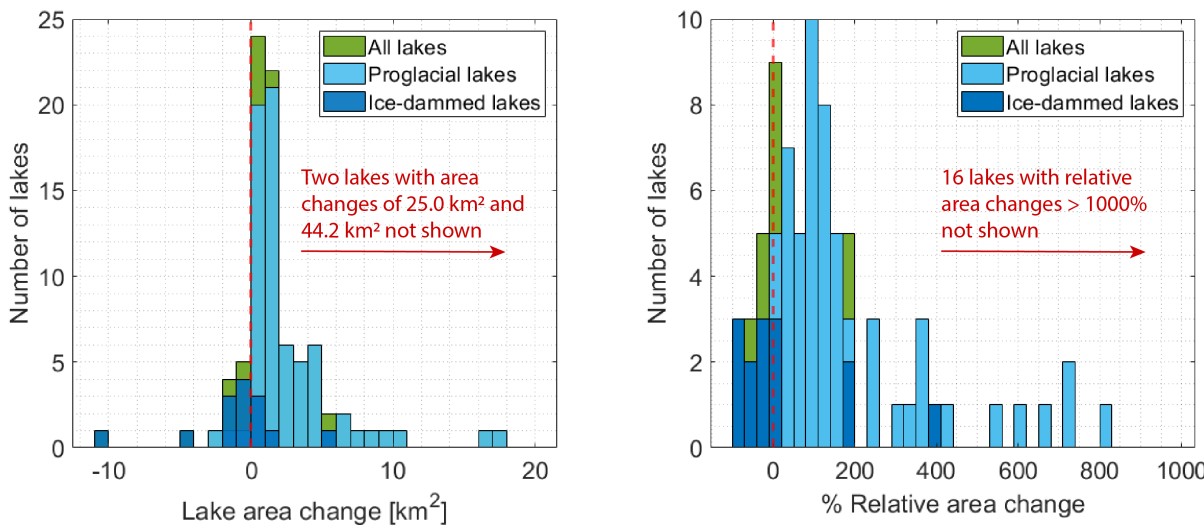

**Figure 4. (left) Distribution of proglacial and ice-dammed absolute area change and (right) relative lake area change.**

**Table 2. Summary statistics for proglacial and ice-dammed study lake area change. Steady lakes are defined as having**
**changed by less than ± 0.1 km². Summary statistics are shown for the change of individual lakes, as well as the cumulative area of all study lakes. For descriptors of individual lakes, we use the robust statistics of the median and 10th and 90th percentile lake area change because the existence of extreme values makes the minimum, mean, and maximum area change less meaningful. Relative area change is scaled by a lake's initial area, so a 100% increase indicates a lake that doubled in area, while -100% indicates a lake that completely disappeared.**

| | *Proglacial* | | | *Ice-dammed* | | |
|---|---|---|---|---|---|---|
| **Number of lakes** | **Growing** | **Steady** | **Shrinking** | **Growing** | **Steady** | **Shrinking** |
| **(- , %)** | 72 (83%) | 11 (13%) | 4 (5%) | 3 (15%) | 8 (40%) | 9 (45%) |
| **Absolute area change** | **10th %** | **Median** | **90th %** | **10th %** | **Median** | **90th %** |
| **(individual, km²)** | 0.01 | 1.28 | 6.76 | -3.7 | -0.04 | 0.36 |
| **Relative area change** | **10th %** | **Median** | **90th %** | **10th %** | **Median** | **90th %** |
| **(individual, %)** | 8% | 125% | >1000% | -82% | -15% | 212% |
| **Cumulative area** | **1984** | **2018** | **Change** | **1984** | **2018** | **Change** |
| **(km²)** | 336 | 606 | 270 (81%) | 96 | 80 | -17 (-17%) |


Of the 107 ice-marginal lakes considered in this study, 17 % detached from their associated glacier during our study period or between the Post and Mayo (1971) catalog and the beginning of our record. Lakes that detached from their associated glacier are found throughout the study region (Fig. 5). Nine proglacial lakes formed during the study period, with no new ice-dammed lakes observed in our lake subset. We use the term "new lakes" to denote lakes that formed during the study period (e.g., Fig. 3), but do not separate these lakes for later statistical analyses. Of growing lakes, 50 lakes (73 %) exhibit linear growth, while 8 (12 %) and 10 (15 %) lakes exhibit accelerating and decelerating growth, respectively. Of shrinking lakes, 9 (75 %) exhibit linear shrinkage, while two (17 %) and one (8 %) lake exhibits accelerating and decelerating shrinkage, respectively (Fig. S4).

There is no obvious spatial organization of observed lake area change (Fig. 5), with all manners of change observed across the study area. We again stress that we investigate a subset of ice-marginal lakes (n = 107), not every lake in the area, and determining their representativeness of population-scale regional lake behavior must be the subject of future work.

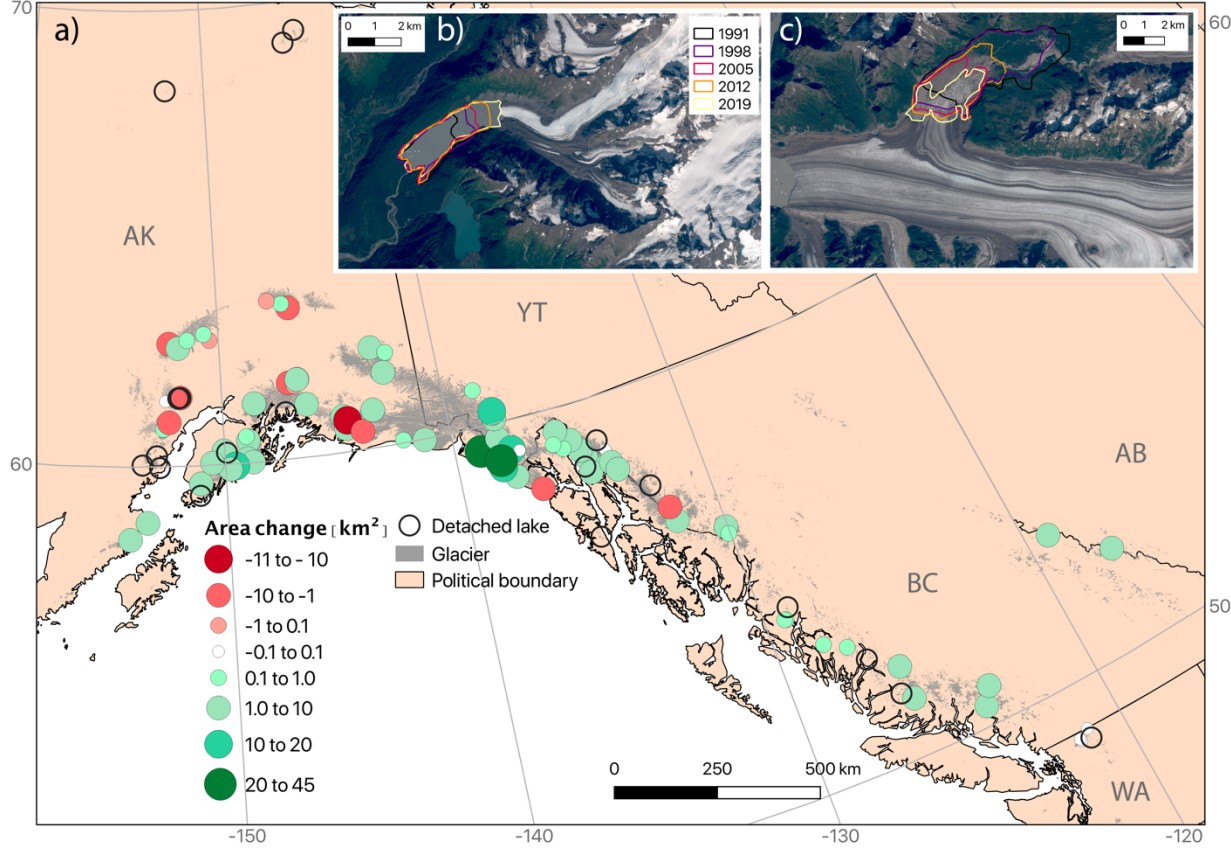

**Figure 5: a) Absolute area change for the studied ice-marginal lakes between 1984 and 2018, including both proglacial and ice-dammed lakes. Green (red) circles indicate lakes that grew (shrunk) over the study period. White circles indicate lakes that remained relatively stable (within ±0.1 km² of their initial area), while unfilled circles show lakes that detached from their associated glacier during the study period. Glacier extent is shown in gray fill (RGI, 2017) and black lines indicate political boundaries. Examples are shown of a b) growing proglacial lake (Unnamed lake downstream from Twentymile**

**Glacier; 60.94 N, -148.78 E) and a c) shrinking ice-dammed lake (Van Cleve Lake dammed by Miles Glacier; 60.70 N, -144.41 E). Years displayed in b) and c) are upper limits on a lake's outline (e.g., a lake delineation between 1991 and 1998 will appear as a purple line). Background imagery in b) and c) is from Landsat 8.**


### 4.2 Bivariate correlations with absolute lake area change

### 4.2.1 Climatic correlations with absolute area change

We investigate the potential influence of climatological variables on absolute ice-marginal lake area change between
1984 – 2018 using the nonparametric Kendall correlation test (Table 3). Average decadal summer (June, July, Aug.) air temperature is positively associated with proglacial lake absolute area change ($p < 0.05$; $\tau = 0.19$) and winter (Dec., Jan., Feb.) precipitation is inversely correlated with ice-dammed lake area change ($p < 0.05$; $\tau = -0.46$). As mentioned in Sec. 2.2, we run correlations between winter precipitation and summer temperature because these are the climate variables most relevant to glacier mass balance. Physically, these correlations mean that proglacial lakes in regions
with warm summers are growing faster, and ice-dammed lakes in regions with wet winters are shrinking more rapidly. Despite these correspondences with mean climate variables, we find little evidence for relationships between lake area change and the long-term change in summer air temperature or winter precipitation. The greatest rates of absolute ice-marginal lake area change are generally occurring in regions with minimal changes in winter precipitation and moderate warming (Fig. S5-S6). We do observe a significant positive relationship between the change in winter
precipitation and proglacial absolute lake area change, yet there is not a clear physical mechanism to explain greater lake expansion in regions with more winter precipitation – we expand upon this idea in Secs. 4.4 and 5.4. A proglacial lake's distance from the open ocean is inversely associated with its absolute area change ($p < 0.05$; $\tau = -0.23$; Fig. S9), indicating that coastal proglacial lakes are growing faster than inland lakes. The strength of this correlation is of similar magnitude to those relating proglacial lakes to other climate variables, and in Sec. 4.3 we argue covariance
between climate variables and continentality provides a more plausible explanation for unintuitive correlations between absolute lake area and climatic variables.

**Table 3. Kendall rank correlation coefficient (τ) values for monotonic relationships between absolute (middle columns) and relative (rightmost columns) lake area change with associated climatological, glaciological, and topographic variables. In each category, test statistics are reported separately for proglacial and ice-dammed lakes. Bold numbers indicate correlations that are significant at $p \leq 0.05$, while regular text indicates relationships where $0.05 < p \leq 0.1$. Dashes indicate a correlation with $p > 0.1$. Positive (negative) correlation coefficients indicate a direct (inverse) relationship between the examined variables.**

| | Parameter | Absolute area change | | Relative area change | |
|---|---|---|---|---|---|
| | | *Proglacial* | *Ice-dammed* | *Proglacial* | *Ice-dammed* |
| **Climatological** | Mean summer temperature (2000s) | **0.19** | - | -0.13 | - |
| | Change in summer temperature (2000s-1960s) | - | - | - | - |
| | Mean winter precipitation (2000s) | - | **-0.46** | - | - |
| | Change in winter precipitation (2000s-1960s) | **0.20** | - | -0.15 | - |
| | Distance to open ocean | **-0.23** | - | 0.16 | |
| **Glaciological** | Glacier area | **0.22** | - | - | - |
| | Glacier width | **0.32** | - | - | - |
| | Median lake-adjacent ice thickness | **0.25** | - | - | 0.47 |
| | Mass balance gradient | - | - | **-0.18** | - |
| | 2010s average annual mass balance | **-0.17** | - | - | - |
| | 1980-2016 summed annual mass balance | - | - | - | - |
| **Topographic** | Latitude | - | - | - | - |
| | Longitude | - | - | - | - |
| | Elevation | **-0.27** | - | **0.19** | - |
| | Initial lake area | **0.33** | **-0.41** | **-0.53** | - |

## 4.2.2 Glaciologic correlations with absolute area change

We find statistical associations between several glaciologic variables and absolute proglacial lake area change, but not with ice-dammed lake area change (Table 3). For all lakes, the only glacier mass balance variable with a statistically significant correlation with absolute lake area change is the average mass balance in the 2010s ($\tau = -0.17$; Fig. S7a). The sign of this correlation indicates that proglacial lakes are growing more rapidly downstream from glaciers with a more negative mass balance in recent times. Notably, we do not find any statistical links between lake area change and the associated glacier's cumulative mass balance over the 1980 – 2016 period (Table 3). Considering glacier geometric factors, however, we find several significant correlations with proglacial lake area change (Table 3). Glacier area ($\tau = 0.22$), width ($\tau = 0.32$; Fig. S8a), and near-terminal median ice thickness ($\tau = 0.25$; Fig. 6) all exhibit correlations with proglacial lake area change at a $p < 0.05$ level (Table 3). This indicates that proglacial lakes are growing most rapidly where they exist downstream of large and wide glaciers with thick ice near the terminus. We

find no evidence for statistical links between absolute ice-dammed lake area change and glacier geometric nor mass
       balance variables (Table 3).

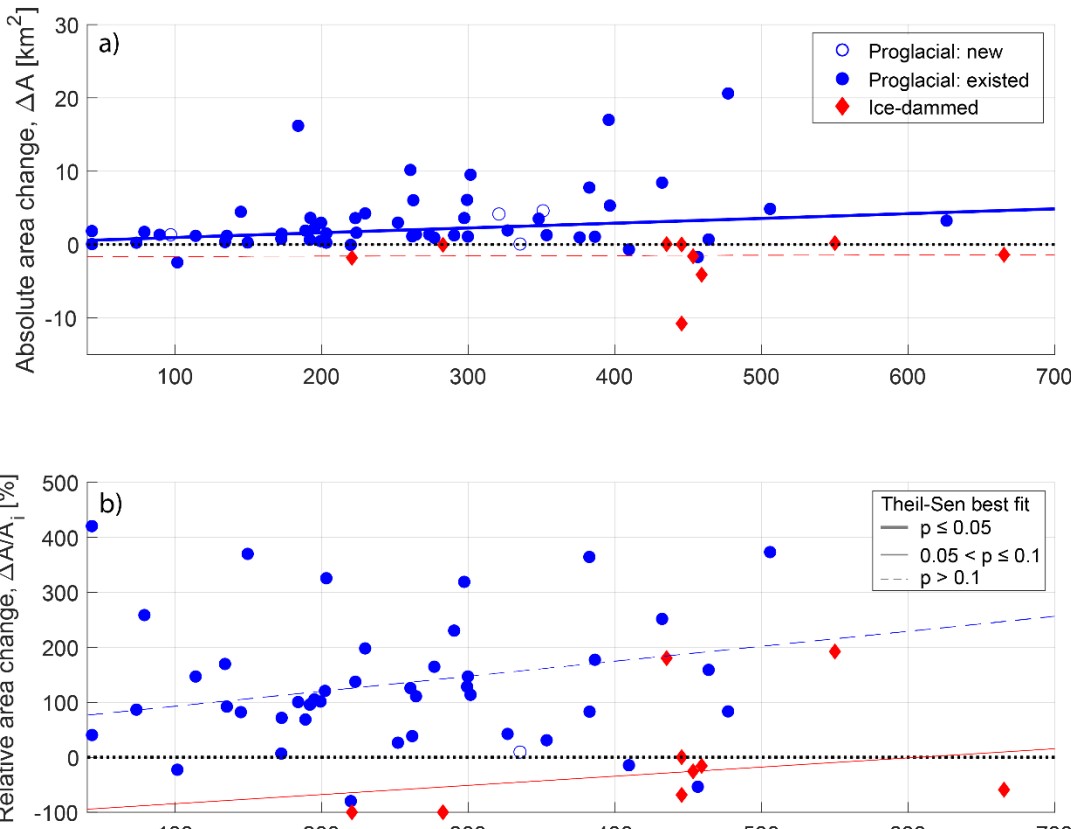

**Figure 6. (a) Absolute and (b) relative lake area change as a function of median lake-adjacent glacier ice thickness (see**
**Section 2.3) for proglacial (blue circle) and ice-dammed (red diamond) lakes. On both panels, lines show the linear fit to**
       **proglacial (blue) and ice-dammed (red) lakes as estimated to by the non-parametric Theil-Sen robust line. Thick solid lines**
       **show relationships that are significant at the $p \leq 0.05$ level, thin solid lines show $0.05 < p \leq 0.1$ relationships, and thin dashed**
       **lines show $p > 0.1$ relationships. All significance values are estimated by the Kendall rank correlation test. The black dotted**
       **line shows zero lake area change. Unfilled symbols indicate lakes that appeared during the study period.**


### 4.2.3 Geometric and geomorphic correlations with absolute area change

       Of all our climatic, glaciologic, and geometric variables, initial lake area is one of the strongest predictors of absolute
       lake area change, exhibiting a moderately strong statistically significant positive association with proglacial lake area
       change ($\tau = 0.33$; Table 3; Fig. 7a) and a strong inverse relationship with ice-dammed lake area change ($\tau = -0.41$;
Table 3; Fig. 7a). We also find that glacier width at terminus ($\tau = 0.32$) is significantly associated with lake area
       change. Additionally, a moderately strong inverse relationship exists between absolute lake area change and elevation
       ($\tau = -0.27$; Fig. 8a), with low elevation lakes growing most rapidly. Together, these associations suggest that large,

low elevation lakes occupying wide valleys have grown most rapidly over the 1984 – 2018 study period. Harlequin Lake (below Yakutat Glacier, Alaska; Fig. 2a), the fastest growing study lake ($\Delta A = 44.2$ km$^2$), exemplifies these traits.

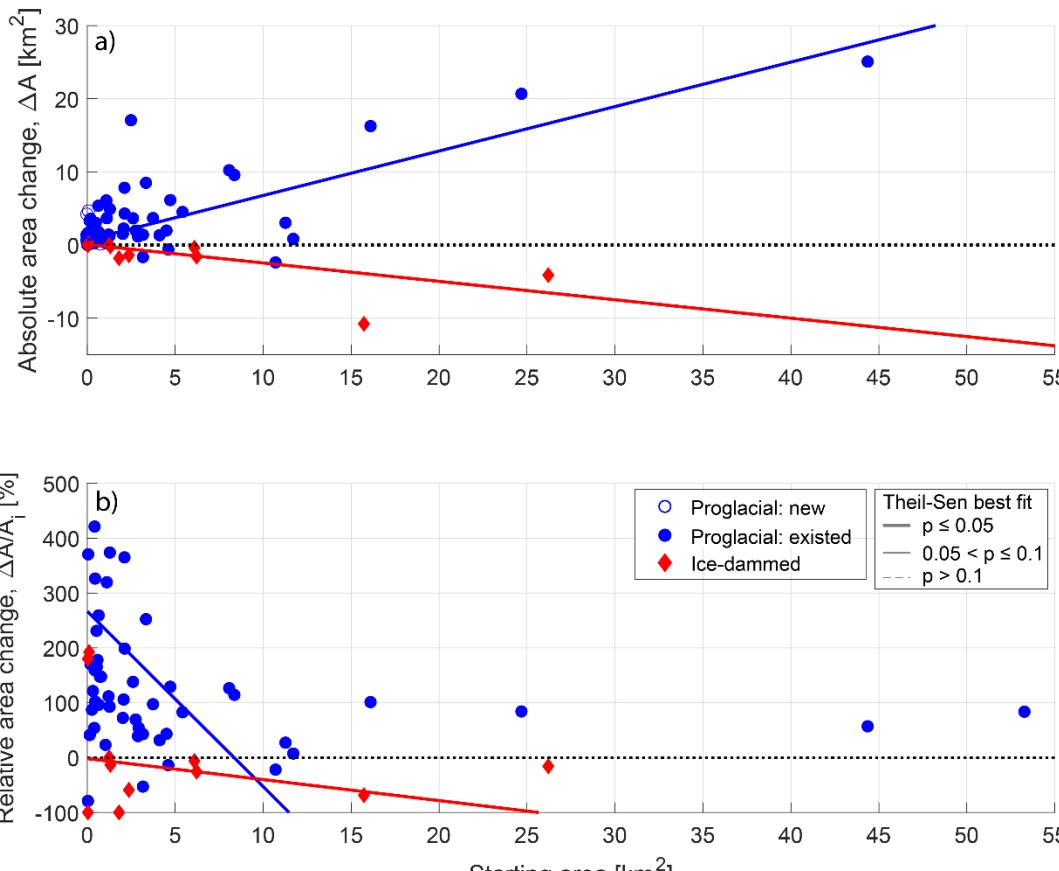

**Figure 7: (a) Absolute lake area change as a function of initial lake area for all proglacial lakes (blue circles) and ice-dammed lakes (red diamonds). (b) Relative lake area change as a function of initial lake area. On both panels, lines show the linear fit to proglacial (blue) and ice-dammed (red) lakes as estimated to by the non-parametric Theil-Sen robust line. Thick solid lines show relationships that are significant at the $p \leq 0.05$ level, thin solid lines show $0.05 < p \leq 0.1$ relationships, and thin dashed lines show $p > 0.1$ relationships. All significance values are estimated by the Kendall rank correlation test. The black dotted line shows zero lake area change.**

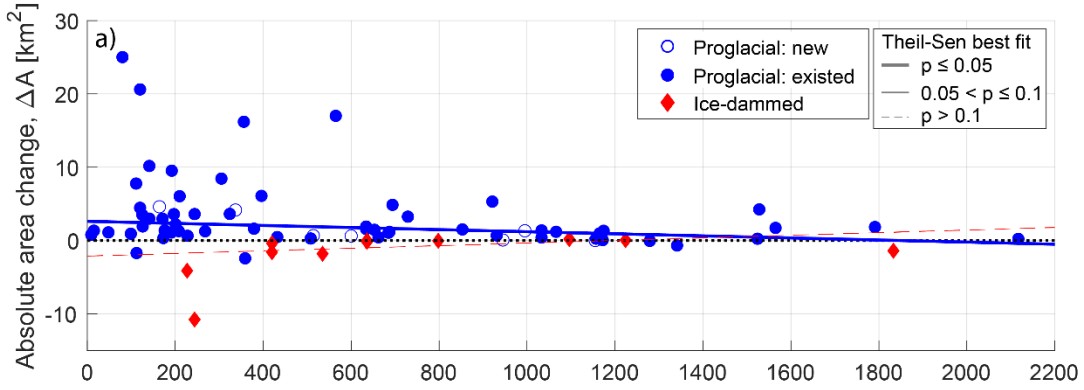

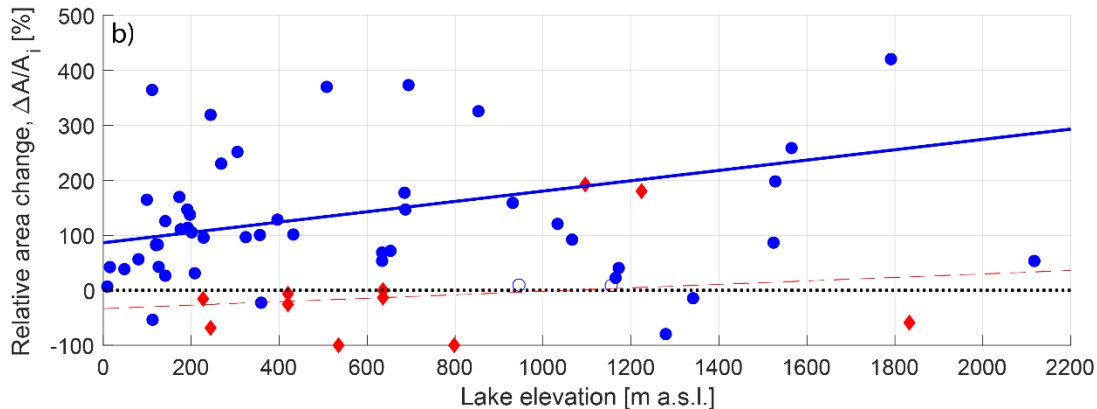


**Figure 8: (a) Absolute and (b) relative lake area change as a function of lake elevation for proglacial (blue circle) and ice-dammed (red diamond) lakes. On both panels, lines show the linear fit to proglacial (blue) and ice-dammed (red) lakes as estimated to by the non-parametric Theil-Sen robust line. Thick solid lines show relationships that are significant at the $p \leq 0.05$ level, thin solid lines show $0.05 < p \leq 0.1$ relationships, and thin dashed lines show $p > 0.1$ relationships. All significance**

**values are estimated by the Kendall rank correlation test. The black dotted line shows zero lake area change. Unfilled symbols indicate lakes that appeared during the study period.**

### 4.3 Bivariate correlations with relative lake area change

In Section 4.2, we discussed statistical associations between environmental variables and *absolute* lake area change.

In this section, we investigate statistical links between *relative* lake area change and those same environmental variables. We first discuss statistical results for climatic variables, followed by glaciologic and geometric variables.

We find no statistically significant links between climate variables and relative ice-dammed lake area change, with a few p < 0.1 associations for proglacial lakes (Table 3). The same climatic variables that were significant for absolute area change are again significant for relative proglacial lake area change, though with opposite signs. We observe

inverse correlations between relative proglacial lake area change and average summer air temperature ($\tau = -0.13$, p

= 0.02) as well as the change in winter precipitation ($\tau = -0.15$, $p = 0.01$). We find a direct relationship between relative proglacial lake area change and distance from the open ocean ($\tau = 0.16$, $p = 0.01$; Figs. 9c and S9b). As we discuss in Secs. 4.2 and 4.4, summer air temperature and winter precipitation change are both themselves correlated with distance from the open ocean (Figs. 9c-10), and we suggest continentality is the most physically-plausible driver of observed statistical links. While maritime proglacial lakes are growing most rapidly in terms of *absolute* area, interior proglacial lakes are growing most rapidly *relative* to their initial size (Fig. S9b).

Relatively few of the considered glaciologic variables are significantly correlated with relative ice-marginal lake area change. However, we do find a strong direct relationship between relative ice-dammed lake area change and lake-adjacent ice thickness ($\tau = 0.47$, $p = 0.07$; Fig. 6b). Physically, this suggests that lakes dammed by thick glaciers have shrunk least, relative to their initial area. Additionally, relative proglacial lake area change is inversely correlated with the associated glacier's mass balance gradient ($\tau = -0.18$, $p = 0.04$; Fig. S7b). This indicates that proglacial lakes downstream from glaciers with "flat" mass balance gradients (i.e., little change in mass balance with increasing elevation) have grown most rapidly, relative to their initial area. This is consistent with interior proglacial lakes growing more rapidly in relative terms, because maritime glaciers generally have steeper mass balance gradients, with the opposite being true for continental glaciers, as discussed in greater detail in Sec. 4.4.

For the geometric and geomorphic variables, we again find the same statistically significant variables as seen for absolute area change, but with opposite sign. While low elevation lakes tend to grow more rapidly in terms of absolute area change, high elevation lakes grow more quickly in relative terms ($\tau = 0.19$, $p = 0.02$; Fig. 8b). We observe a strong inverse correlation between relative lake area change and initial lake area ($\tau = -0.52$, $p < 0.01$; Fig. 7b), but we interpret this to be an artifact of data processing because initial lake area is used to compute relative lake area change. That being said, this result suggests that smaller lakes are experiencing greater relative area change, while large lakes are experiencing greater absolute change.

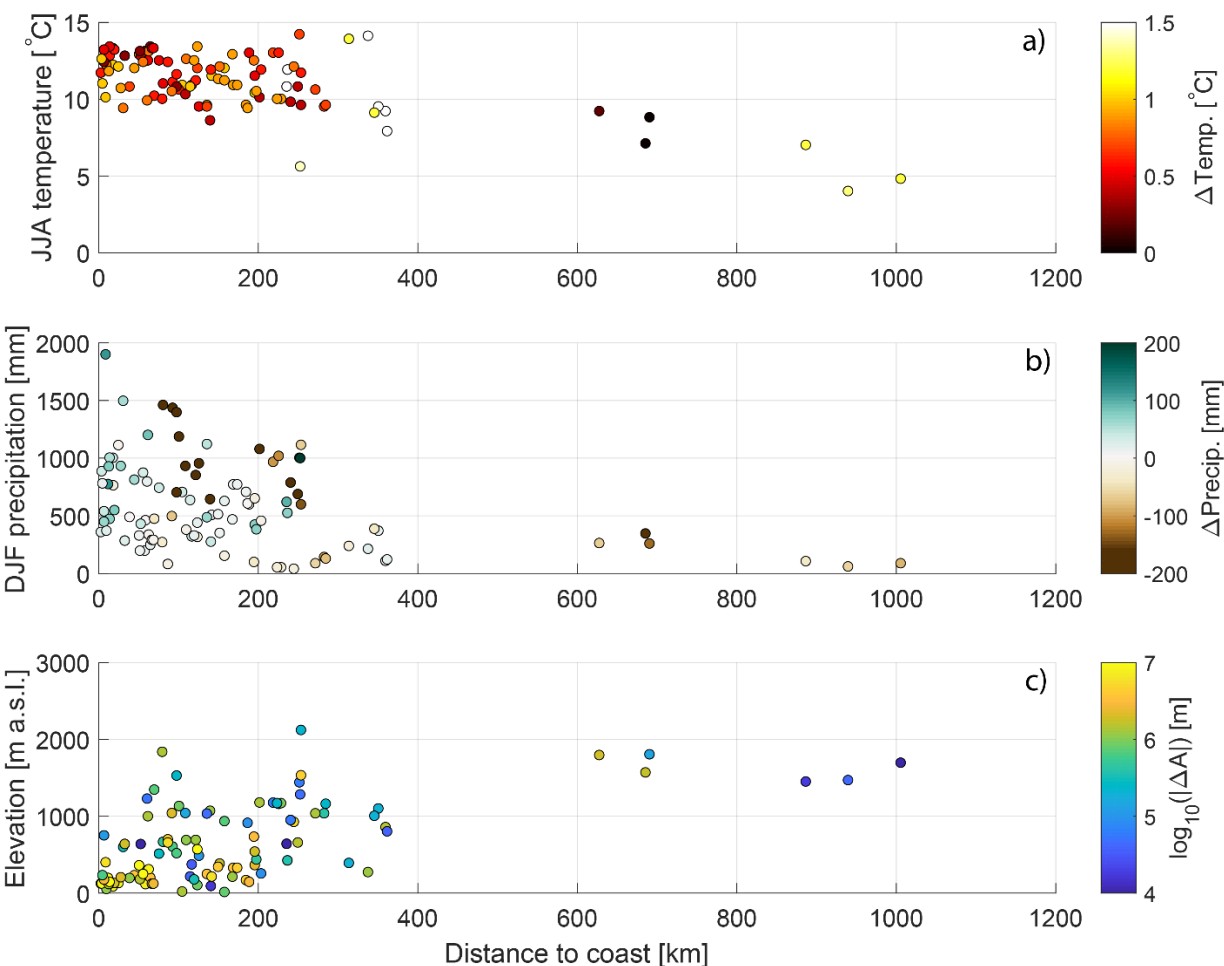

**Figure 9. Variation in climatic and topographic variables as a function of a lake's distance from the open ocean. (a) Summer air temperature (y axis) and its change (colors) between the 1960s and 2000s. (b) Winter precipitation (y axis) and its change between the 1960s and 2000s. (c) Lake elevation (y axis) and absolute lake area change between the 1984 and 2018.**

## 4.4 Assessing covariance of environmental variables

It is plausible that a correlation between lake area change and a single environmental variable is actually due to underlying covariance amongst the environmental variables. Covariance between environmental variables in some ways complicates interpretation of the results presented in Secs. 4.2-4.3, but this covariance also provides a physically plausible explanation for several unintuitive results presented in that section. We cross-correlate the 15 environmental variables shown on Table 3, and find that most (63 %) of the possible pairs of environmental variables are significantly correlated at $p < 0.05$ (Fig. 10; Table S4). These correlations signify that one environmental variable (e.g., summer temperature) systematically varies with another (e.g., latitude), which is driven by the spatially coherent structuring of these variables. Below, we describe several salient clusters of correlated environmental variables that affect interpretation of results presented in Sections 4.2-4.3.

Lake elevation and initial lake area are both significantly correlated with 50 % of the other environmental variables, and the variables with which they covary are nearly identical (Fig. 10; Table S4). Large, low-elevation lakes are

significantly associated with the following variables: proximity to the coast, high summer temperatures, winters that have gotten wetter, larger glaciers, wider glaciers, thicker glaciers, and glaciers with a steeper mass balance gradient. Distance to the coast is significantly correlated with a similar set of environmental variables, but lacks significant association with the variables describing glacier size (i.e., area, width, lake-adjacent ice thickness).

Notably, variables describing glacier mass balance are not significantly correlated with lake elevation, initial area, or
distance from the coast. Glaciers with more negative cumulative mass balance instead are significantly associated with locations further south and east; warm summers; wet winters; winters that are becoming drier; smaller, narrower, and thinner glaciers, and; glaciers with a steep mass balance gradient.

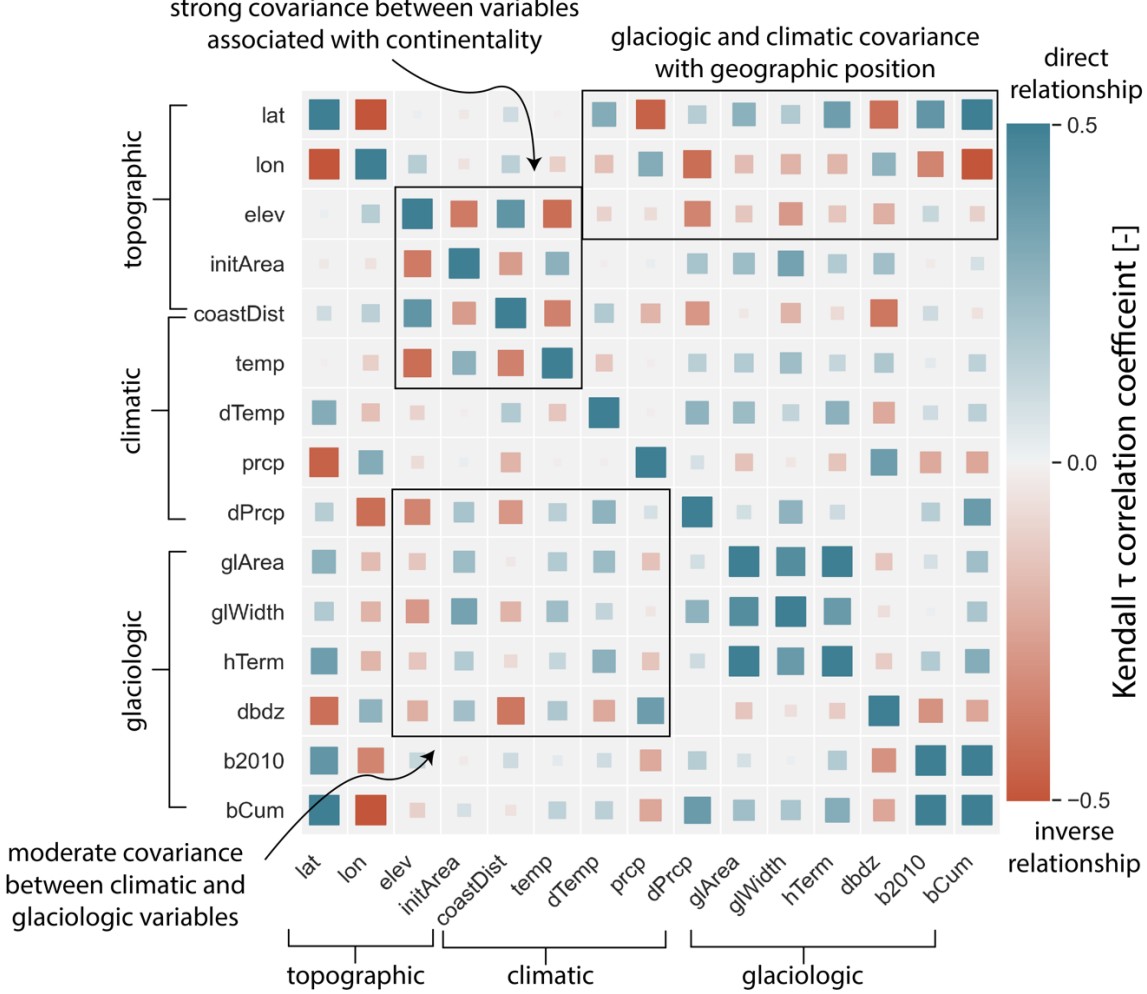

**Figure 10. Correlation matrix showing interrelatedness of the topographic, climatic, and glaciologic variables used for lake area change correlations (Table 2). Symbol color scales with the Kendall tau correlation coefficient between the environmental variables associated with that point's row and column. Symbol size scales with the absolute value of the correlation coefficient. Black boxes highlight covarying environmental variables that are discussed in the text. Meanings for variable names are as follows: lat = latitude; lon = longitude; elev = lake elevation; initArea = lake initial area; coastDist = distance from the open ocean; temp = JJA temperature; dTemp = change in JJA temperature; prcp = DJF precipitation; dPrcp = change in DJF precipitation; glArea = area of the lake-adjacent glacier; glWidth = width of that glacier; hTerm = lake adjacent median ice thickness; dbdz = mass balance gradient; b2010 = average annual mass balance for 2010 – 2016; bCum = cumulative mass balance for 1980 – 2016.**

### 4.5 Multivariate correlations between lake area change and environmental variables

To supplement bivariate correlations and the above discussion of covariance between environmental variables, we undertake non-parametric correlation testing between ice-marginal lake area change and principal components scores. We first must interpret the physical meanings of the PC axes. The first four principal components axes explain 73.3 % of the variance found in the 15 environmental variables for which we present correlation results (Table 2). After inspecting the loadings of environmental variables onto each principal component axis (Table S2) and plotting the

quasi-exponential decay explained by increasing PC axis numbers, we retain only the first four PC axes for correlation testing. Some interpretation is required to understand the physical meaning of PCA results: Strong loadings of PC axis (PC1, absolute value $\geq 0.25$) are found for environmental variables associated with a lake's geographic location (e.g., latitude, longitude, elevation), and so we interpret this axis to largely reflect a lake's position on the earth surface. This interpretation is supported by PC scores varying systematically along a latitudinal gradient (Fig. S3a). Strong

PC2 loadings are indicative of a maritime setting; high PC2 scores are associated with low elevation, proximity to the open ocean, high mass balance gradients, and high winter precipitation (Table S2). This interpretation is again supported by the spatial distribution of PC2 scores, with a systematic decrease in scores moving away from the coast (Fig. S3b). Glacier-size related variables load most strongly onto PC3, while climate and climate change-related variables load most strongly on PC4 (Table S2; Fig. S3-d). We thus interpret PC axes 1 – 4 to aggregate individual

environmental variables related to geographic position (i.e., latitude and longitude), continentality, glacier size, and climate (and its change), respectively.

The results show significant correlations for only PC2 (Table 4), the PC axis we interpret to reflect continentality (Sec 3.4). Proglacial (ice-marginal) lakes exhibit a significant direct (inverse) correlation between PC2 and absolute lake

area change. An inverse correlation ($p \leq 0.05$) exists between PC2 and relative proglacial lake area change (Table 4). Due to the details of PCA data transformation, high PC2 scores are related to factors associated with a maritime setting. Therefore, a positive correlation coefficient indicates greater lake growth being associated with a maritime setting. A negative correlation coefficient either indicates greater lake area decline in a maritime setting, or greater area increase towards the continental interior. These findings are consistent with the bivariate correlation results (Sec.

4.2 – 4.3; Table 3), in which coastal proglacial (ice-dammed) lakes experienced the highest rates of absolute area increase (decrease), while interior proglacial lakes experienced the higher rates of relative area change.

In addition to these results, the relative area change of proglacial lakes is directly correlated ($0.05 < p \leq 0.1$) with PC3 (Table 4), which we interpret to reflect glacier size (Sec 3.4). This suggests that proglacial lakes associated with

large glaciers are experiencing higher rates of relative area change. No significant relationships between glacier geometric characteristics and relative area change were found for the bivariate correlations (Table 2). This discrepancy in significance between single-variable and multivariate analyses suggests that either: 1) by combining multiple glacier-related characteristics, the association between glacier size and relative lake area change becomes more apparent; 2) other variables that load strongly onto PC3 (e.g., elevation, longitude) underly this association, or;

3) our interpretation of the physical meaning of PC3 is incorrect. No other PC scores were associated with ice-marginal lake area change, either in an absolute or relative sense at the $p \leq 0.1$ level (Table 4).

**Table 4. Kendall rank correlation coefficient (τ) values for monotonic relationships between absolute and relative lake area change with the four leading principal components axis scores. The interpretation for a physical meaning of each axis is listed beside the axis number. In each category, test statistics are reported separately for proglacial and ice-dammed lakes. Bold numbers indicate correlations that are significant at *p* ≤ 0.05, while regular text indicates relationships where 0.05 < *p* ≤ 0.1. Dashes indicate a correlation with *p* > 0.1. Due to details of PCA data transformation, a positive (negative) correlation with PC2 indicates higher proglacial (ice-dammed) lake area growth (shrinkage) being associated with environmental variables characteristic of a maritime setting (e.g., close to coast, high mass balance gradient, low elevation). A positive correlation with PC3 indicates higher proglacial lake growth being associated with variables characteristic of larger glaciers (e.g., high glacier area, high lake adjacent ice thickness).**

| PC axis number | Interpretation | Absolute area change | | Relative area change | |
|---|---|---|---|---|---|
| | | *Proglacial* | *Ice-dammed* | *Proglacial* | *Ice-dammed* |
| 1 | Spatial location | - | - | - | - |
| 2 | Continentality | **0.31** | **-0.61** | **-0.22** | - |
| 3 | Glacier size | - | - | 0.18 | - |
| 4 | Climate (change) | - | - | - | - |

## 5 Discussion

The discussion aims to (1) put our findings of regional lake area change behavior in context with global ice-marginal lake change found in earlier works; (2) interpret the physical meaning of the pattern of statistical associations between predictor variables and absolute and relative lake area change, and; (3) examine the limitations of our datasets and our analyses.

### 5.1 Regional lake change behavior

We observe diverging trends in lake area between studied ice-dammed and proglacial lakes. Many ice-dammed lakes (45 %) are shrinking in absolute area, while most proglacial lakes (83 %) are growing (Figs. 3-4; Table 2) and proglacial lakes also increase in number. This dichotomy makes intuitive sense in the context of widespread glacier wastage in this area (Arendt et al., 2009). Proglacial lakes expand headward as their associated glaciers retreat. Meanwhile, ice-dammed lakes shrink because thinner ice dams are less capable of impounding large reservoirs, and ice-dammed tributary valleys are drained as trunk glaciers retreat. We find an average area decrease of 17 % among our studied ice-dammed lakes, slightly lower but broadly similar to the estimates of Wolfe et al. (2014), who found a 28 % decrease in Alaska ice-dammed lake area between 1971 – 2000. We note that, while we here document lake area change, i.e. a readily observable quantity, similar results would likely emerge if we estimated lake volume change because lake area scales with lake volume (Cook and Quincey, 2015; Shugar et al., 2020). However, converting area to volume in the absence of field observations requires the use of empirical scaling relationships (Cook and Quincey, 2015; Shugar et al., 2020) and would make the presented data more uncertain; we therefore only consider area change in the present study.

Similar studies of proglacial lakes undertaken across the Himalayas (Gardelle et al., 2011; Shukla et al., 2018; Wang et al., 2015; Zhang et al., 2019), northern Europe (Canas et al., 2015; Tweed & Carrivick, 2015), and Andes (Wilson et al., 2018; Emmer et al., 2020) found increases in proglacial lake area ranging from 7 % to 110 %. We find that between 1984 – 2018 proglacial lakes in northwestern North America investigated in this study have increased in cumulative areal coverage by approximately 58 %, with a median individual lake growth of 125 % (1.28 km$^2$). In aggregate, this increase in proglacial lake area is also in agreement with conceptual models of proglacial lake expansion in size and number as overdeepened basins are exposed as their upstream glaciers retreat (Emmer et al., 2020; Otto, 2019). The fact that our aggregate Gulf of Alaska lake area change sits in the middle of previously reported values likely partly stems from the fact that Alaska lakes are the early to middle stages of proglacial lake development. In less heavily glaciated areas such as the Peruvian Cordillera Blanca or European Alps, the extant glaciers have already retreated into steep, high elevation basins with little potential for further lake development, while glaciers in Alaska still extend to flatter low elevations with more potential for lake development following glacier retreat. The extensive debris cover found on Alaska glaciers, most similar to Himalayan glaciers, could affect ice-marginal lake formation in several competing ways. While debris-covered glaciers tend to thin rather than retreat in response to climate warming (potentially limiting lake growth), they are also associated with lower surface slopes (potentially enhancing lake growth; e.g., Anderson et al., 2018). Assessing the importance of such factors provides an avenue for future research. Beyond these physical factors, some variation in lake area change between regions arises from slightly varying temporal spans or definition of glacier-related study lakes.

Lake area change occurs either along a continuum (e.g., a small lake getting bigger) or as a system switch (e.g., lake completely disappearing). These different modes of area change impact their adjoining environments in different ways. We document the temporal growth style of lakes moving along a continuum (Fig. S4) and find the majority of lakes (64 %) exhibit steady, linear growth trends over the study period. Assuming lake area change is tied to glacier retreat, this implies constant rates of glacier retreat, despite generally accelerating rates of mass loss (Gardner et al., 2013; Hugonnet et al., 2021; Zemp et al., 2019). This growth style could reflect the linear planform shape of many valleys in which ice-marginal lakes form, which allow lakes to grow in length but inhibit large changes in width. Of the investigated proglacial lakes (n = 73), ten (14 %) exhibit decelerating change (either growth or shrinkage), which is indicative of either: 1) lake area coming into a steady state in equilibrium with the current environment, or; 2) lakes reaching late stage in their growth history in which they will soon detach from their associated glacier (Emmer et al., 2020). Regardless of the mechanism for decelerating change, both of these styles represent stabilizing lake area. In contrast, eight (11 %) lakes exhibit accelerating change. The paucity of lakes exhibiting stabilizing growth styles suggests that ice-marginal lakes in this area are in the middle stages of their growth history and will likely continue to change for the foreseeable future. Of our 107 study lakes, nine appeared during our study period and 18 disconnected from their associated glacier (three disconnected during our 1984 – 2018 study period, while 15 disconnected before 1984). Either of these transitions mark a fundamental shift in landscape connectivity and function (e.g., Dorava and Milner, 2000; Baker et al., 2016).

The evolution of ice-marginal lakes impacts downstream flood hazard due to their association with glacial lake outburst floods (GLOFs), also known as jökulhlaups. The majority our ice-dammed study lakes shrunk, while proglacial lakes predominantly grew. Maximum outburst flood discharge (both instantaneous and cumulative) scales with the reservoir size (Björnsson, 2010; Nye, 1976). The diverging trends between ice-dammed and proglacial lakes suggests that the outburst hazard associated with ice-dammed lakes may be, on average, decreasing across the study reach, while the hazard associated with proglacial lakes may be growing.

**5.2 Topographic and geometric factors most strongly associated with ice-marginal lake area change**

Both bivariate and multivariate statistical analyses suggest that topographic and geometric controls such as lake elevation and initial area exert the strongest influence on absolute ice-marginal lake area change (Tables 3-4). As we discuss below, even variables we have previously called climatic or glaciologic may be thought of as topographic variables because they are closely associated with the shape of the basin into which a lake may grow as its associated glacier retreats and thins.

Initial lake area is the strongest bivariate predictor for absolute proglacial lake area change ($\tau = 0.33$; Fig. 7a) and is the second strongest predictor for absolute ice-dammed lake area change ($\tau = -0.41$; Fig. 7a). The greatest possible area loss of an ice-dammed lake is that associated with complete lake drainage. Thus, a small ice-dammed lake is fundamentally limited in its maximum area loss, while a large lake can experience significant shrinkage. We posit that this geometric control underlies the inverse correlation between absolute ice-dammed lake area change and its initial area. We hypothesize two mechanisms that may explain the fact that initially larger proglacial lakes have grown faster than initially small lakes: 1) The initial existence of a large lake requires a large basin, and basins generally do not end abruptly. Therefore, the simple existence of a large lake suggests that there is higher potential growth in a regionally-extensive depression. This explanation would require Alaska's proglacial lakes to be in an early stage of development (Emmer et al., 2020), with ample room to grow into overdeepened basins. Alternatively, 2) larger lakes likely have greater surface area at the glacier-lake interface, which may lead to higher rates of frontal ablation. Simply, a wider calving front would give rise to greater lake area growth for a set amount of up-valley glacier retreat – a notion supported by our observation that proglacial lakes downstream from wide glaciers have grown most rapidly in absolute terms (Table 3). One can posit other mechanisms to explain this observation, perhaps that large lakes tend to be warmer (Sugiyama et al., 2016; Truffer & Motyka, 2016), which could affect rates of subaqueous melting and, consequently, glacier retreat. Alternatively, lake depth scales with lake area (S. J. Cook & Quincey, 2015), and deeper water at a glacier's terminus generally enhances its calving flux and thus retreat rate (e.g., Benn et al., 2007). Exploring such possibilities provides an interesting opportunity for future research, but is beyond the scope and data constraints of the current study.

Several other factors that are statistically significantly linked can be explained using the framework of topographic factors exerting primary control on absolute lake area change. Lake elevation is inversely associated with absolute proglacial lake area change, with low-elevation lakes growing most rapidly (Fig. 8a; Table 3). A lake's distance to the

ocean may be used to predict absolute proglacial lake area change, with maritime lakes growing most rapidly (Figs. 9 and S9; Table 3). Finally, the median thickness of glacier ice in the region immediately abutting a proglacial lake is directly correlated with that lake's area change, with lakes downstream from thick glaciers growing most rapidly (Fig. 6; Table 3). Multivariate statistics support these interpretations, with significant correlations found between both proglacial and ice-dammed lakes and PC2 scores (Table 4), the PC axis with strong loadings from topographic variables such as lake elevation and distance from the coast (Table S2). All of these associations can be explained by the lake basin geometry expected to be encountered on an idealized transect from the coast towards the interior of the continent, as follows. The Gulf of Alaska region is tectonically active, featured widespread glacier coverage during the Pleistocene (Kauman & Manley, 2004) and has experienced vigorous geomorphic work by glaciers, rivers, and waves (Péwé, 1975). These facts mean that, moving inland from the Gulf of Alaska coast, one first encounters broad lowlands composed of unconsolidated sediment, followed by wide valleys carved by Pleistocene ice streams which have been reworked by modern fluvial processes, and then higher, steeper, and narrow valleys occupied by modern glaciers (Péwé, 1975). In this idealized transect, we expect the large glaciers extending into the coastal plain to be capable of excavating deep basins into weak sediments without significant lateral constraint. Moving inland, steeper and more confined valley geometries inhibit absolute lake growth. Thus, we propose that even variables that at first appear to be associated with climate or glaciology, such as distance from the open ocean or glacier area, may actually be associated with absolute lake area change due to underlying links with lake basin geometry.

In contrast, several of the same climatic, glaciologic, and topographic variables discussed above for *absolute* lake area change exhibit statistically significant relationships with relative proglacial lake area change, but with the opposite sign. In terms of *relative* area change, it is the inland, high elevation proglacial lakes that are growing most rapidly. This finding is consistent with the global-scale study of Shugar et al. (2020), who observed that the increase in the number of ice-marginal lakes primarily occurred through the generation of new lakes at high elevation. Like that work, our results suggest that inland, high-elevation regions are undergoing greater relative change, which is especially relevant given the potential for hydropower development in these locations (Farinotti et al., 2019b).

**5.3 Lack of evidence for strong direct climatic or glaciologic association with ice-marginal lake area change**

Beyond the variables discussed above, which we argue largely reflect lake-adjacent topography, bivariate and multivariate statistical analyses suggest that climatic and glaciologic variables exert minimal influence on either absolute or relative ice-marginal lake area change. Though we observe some associations between mean climate and ice marginal lake area change, we do not find any statistically significant associations between temperature change over 1960s – 2000s, and only an unintuitive inverse correlation between winter precipitation change and proglacial lake area change (Secs. 4.2.1 and 5.4; Table 3). We find a $0.05 < p \le 0.1$ relationship between relative proglacial lake area change and PC3 (Table 4), the PC axis with strong loadings from variables associated with glacier size (Sec. 3.4). Aside from this weaker correlation, though, we find no multivariate statistical evidence for associations between lake area change and the PC axes that load strongly with glaciological or climatological variables (Table 4). This is somewhat surprising, because glacier change must somehow be linked to ice-marginal lake area change, and glacier

change is sensitive to these quantities. Where we do find statistically significant relationships between absolute lake area change and climatic factors, they occur in manners that defy simple physical explanation. For example, it is difficult to see why proglacial lakes experiencing positive/neutral changes in winter precipitation would experience greater lake growth (Figs. S5, S6d, S10d; Table 3) because increasing winter precipitation likely benefits glacier mass balance and would thus inhibit glacier retreat and associated lake growth. We suggest these correlations with climatic

factors reflect underlying covariance in our datasets, discussed in greater detail in Sections 4.4 and 5.4. Summer temperature, winter precipitation, and their changes, all systematically vary with distance from the open ocean (Figs. 9-10), as does a lake's elevation (Fig. 10) and we suggest seemingly unintuitive climatic correlations, such as that described above, are actually driven by a lake's distance from the coast (Figs. 9, 10, and S9; Table S4). Lakes that are further inland experience a more continental climate, but this relationship could also largely express topographic and

geometric controls, as described in Section 5.2. We note that we do not run correlations between lake area change and mean annual precipitation because variations in winter precipitation and summer temperature show strong relationships with Alaska glacier mass balance, particularly for coastal glaciers (e.g,. McGrath et al., 2017), though changes in precipitation throughout the whole year could be more important for glacier mass balance elsewhere. Probing relationships with environmental variables beyond those presented here provide productive avenues for future

research. The lack of strong associations with these external factors may suggest that although climate and associated glacier change are the overarching factors of lake area change, the specific response of a lake to these changes is largely shaped by local factors, such as overdeepening shape and associated lake growth potential. These local factors are more closely tied to topography than climatological or glaciological factors.

Another reason that we do not observe strong associations between climatic factors with ice-marginal lake area change could be due to the processes underlying glacier evolution obscuring the climate signal. Glaciers display varied sensitivity to climatic forcing (e.g., Jiskoot et al., 2009; McGrath et al., 2017; O'Neel et al., 2019; McNeil et al., 2020), so we may expect glacier mass balance, rather than climatic factors alone, to better explain lake behavior. We do find an association between absolute proglacial lake area change and average annual glacier mass balance over 2010 –

2016 (Table 3; Fig. S7a), but do not observe links with decadal average mass balance for any other period, nor cumulative mass balance over a longer period. The variable sensitivity of glacier length change to mass balance perturbation (e.g., Che et al., 2017) likely complicates the link between lake area change and glacier mass balance. We therefore suggest the lack of a statistical relationship between most mass balance variables and lake area change is due to the fact that glacier retreat, and associated lake growth, is responding to climate change in a lagged and

smoothed manner. The average response time of lake-associated glaciers is 92 years (Fig. S1d), while our record length is only 34 years. Thus, the relevant period of climate change to best predict lake area change may either require a longer or earlier period of record than we investigate.

**5.4 Data and statistical limitations**

When considering climatological, glaciologic, and topographic controls on lake area change, it is important to note that these variables are often intertwined (Fig. 10; Table S4). For example, glacier thickness, area, and slope are highly

correlated (e.g., Bahr et al., 2015). Further, we expect these glaciologic variables to be related to climate – a large glacier is more likely to be found in an area of high winter precipitation and low summer temperature. We provide this as one example of interrelated environmental variables, but acknowledge that the existence of covariance between environmental variables is pervasive in this dataset and makes interpretation of statistical results more complicated. Principal components analysis does not entirely solve this problem because, while it provides independent PC axes, an individual axis still contains correlated environmental variables, and thus does not allow for the disentanglement of all variables. In previous sections, we present several statistical associations that lack a straightforward physical explanation and suggest that underlying covariance in environmental variables is the most plausible explanation for these associations. Proximity to the coast and general topographic features associated with coastal or interior settings provides a coherent and physically plausible framework for understanding many of these unintuitive correlations. However, a lake's distance from the coast is also correlated with its initial area and elevation (Fig. 10). This covariance means that we cannot definitively say whether it is a lake's proximity to the coast, its size, or elevation is what truly matters most in providing a physical explanation for the observed patterns of ice-marginal lake area change. However, highlighting the fact that these correlations exist provides testable hypotheses for future studies to investigate the physical mechanisms underlying these relationships in more detail.

We find more statistically significant associations between climatic, glaciologic, and topographic variables and proglacial lake area change than we do for ice-marginal lake area change. This may occur because proglacial lakes are actually more sensitive to these environmental factors, but there is likely some role due to differing sample sizes between the proglacial and ice-dammed groups. Due to exclusion of lakes that detach from their associated glacier during the study period, our statistical analyses investigate 73 proglacial lakes but only 14 ice-dammed lakes. For a given effect size (i.e., correlation strength), a smaller sample will produce a higher p-value (i.e., less significant) than a larger sample (Helsel and Hirsch, 1992). Therefore, the fact we observe fewer statistically significant relationships for ice-dammed lakes should not be taken to mean that these relationships do not exist, but simply that a larger scale study is needed to more definitively investigate controls on ice-dammed lake area change. Such a study is beyond the scope of this work.

Additionally, our study of physical controls on lake area change is only as robust as the datasets upon which we rely. In reality, there may be a link between a variable we have investigated and lake area change, but the relationship does not manifest itself in our study because our representation of that variable is in error. These datasets we employ were optimized to minimize misfit over a large area, and the accuracy of a single value for any one glacier or pixel may be higher than the average values reported in Secs. 2.2-2.4. Despite this uncertainty, these datasets provide our best estimates of these values over our study area, which is too large and remote to allow more detailed characterization of individual sites. We therefore utilize these datasets to allow a preliminary investigation of the importance of these factors over a large area, which can later be refined with more detailed studies.

## 6 Conclusion

We investigate the time evolution of 107 ice-marginal lakes across northwestern North America over 1984 – 2018 and find the majority (83 %) of proglacial lakes grew (median relative change = 125 %) while many (45 %) ice-dammed lakes shrunk (median relative change = –15 %). Non-parametric bivariate and multivariate statistical analyses assess correlations between ice-marginal lake area change and potential physical controls such as climatic, glaciologic, and topographic attributes of the regions surrounding each lake. Our findings indicate that factors associated with a lake's geometry and its adjacent topography are most strongly linked to lake area change. Large, coastal, low-elevation lakes associated with large, wide, thick glaciers underwent the largest area changes, while small, inland, high elevation lakes changed most relative to their initial areas. Covariance between continentality and climatic variables likely underlies the observed unintuitive correlations with those factors, though this same covariance also makes it difficult to assess whether a lake's distance from the open ocean, its elevation, or its initial area is the most important variable in predicting recent lake area change. We caution authors of similar work to consider such covariance between climatic, glaciologic, and topographic factors when investigating apparent physical controls on lake behavior. We find some evidence for enhanced lake area change being associated with glaciers undergoing greater rates of mass loss over the most recent decade, but do not find correlations with long-term cumulative mass balance nor changes in climatic variables in ways that decrease glacier mass balance (i.e., summer warming, winter drying). We suggest that, while climate change and associated glacier wastage must be the primary external driver for lake area change, topographic and geometric factors exert primary control because a lake cannot expand if no basin exists to accommodate its growth. We have shown that ice-marginal lakes have changed substantially over the Landsat record and that many will likely continue to evolve. These shifts in lake area have likely impacted adjacent biophysical systems by changing the timing and magnitude of water and sediment fluxes and will continue to do so. Our study provides initial suggestions of the environmental variables most strongly associated with ice-marginal lake area change. However, to better understand how these glacial lakes will continue to evolve in the face of global climate change, we must further investigate the physical mechanisms by which ice-marginal lakes change, undertake more sophisticated multivariate analyses of these systems, and explore the influence of environmental factors not examined in this work.

**Code and data availability**

A shapefile of time-varying lake outlines can be found at https://arcticdata.io/catalog/view/urn%3Auuid%3Aaa9c6897-9acb-4f9f-8031-e267043d49ad. Python and Matlab scripts for GIS and data processing as well as statistical analyses can be found at https://github.com/armstrwa/proglacialLakes. Climate reanalysis data are available at http://ckan.snap.uaf.edu/dataset. The Randolph Glacier Inventory is located at https://www.glims.org/RGI/. Geotiffs of ice thickness data from Farinotti et al. (2019) can be downloaded from https://doi.org/10.3929/ethz-b-000315707. Glacier mass balance data are from Huss and Hock (2015) and may be requested from those authors.

**Author contributions**

HRF undertook lake delineation, data processing, statistical analyses, and drafted manuscript text. WHA designed the study, performed geospatial data extraction, advised HRF, secured funding, and contributed to manuscript writing and revision. MH modeled glacier mass balance variables, assisted with statistical design, and contributed to the text.

**Acknowledgements**

We thank Regine Hock for discussing unpublished modeled glacier mass balance data. We appreciate conversations with Dominik Schneider, Steve Hageman, Maggie Sugg, and Hasthika Rupasinghe, which provided great insight on the use of multivariate statistical methods. We thank Dan McGrath for discussing climate reanalysis data and Leif Anderson for discussing glacier response times. We gratefully acknowledge the Department of Geological and Environmental Sciences at Appalachian State University (ASU) for funding HRF as an undergraduate research assistant. We recognize the ASU Office of Student Research for conference travel funding. This work was supported by NSF award OPP-1821002. We thank Jenna Sutherland, an anonymous reviewer, and Associate Editor Chris Stokes for constructive reviews that improved the rigor, clarity, and contextualization of this work.

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
