# Peer review of "Gulf of Alaska ice-marginal lake area change over the Landsat record and potential physical controls"

_The Cryosphere, 2020_

## Referee Comment (RC2)

**1 General Comments**

Dear Editor,

Thank you for the opportunity to review this paper. The authors leverage a record of Landsat imagery to examine the evolution of ice-marginal lakes over the last 3 decades for the Gulf of Alaska. Changes to lake size are compared with available data about glacier dynamics, melt, glacier morphology, topographic conditions and climate data to better understand possible drivers of changes to glacier lake size.

The paper covers important topics that must be approached and I commend the authors for pursuing this original research. Clearly a large amount of novel data was ascertained and analyzed in the context of existing datasets. However, I found some substantial issues with the text which either require clarification or methods need to be adjusted.

In my opinion, the methods and conclusions are generally well-explained. However, I am skeptical of the conclusion that topography is the primary control on ice marginal lake change, given that the authors point to the interaction of topography on climate and glaciological parameters. This seems like a bit of circular argument. Successfully arguing this case will require evaluating the interaction between climate, topography and glaciology and the result deemed independent (or not). If, as stated in the discussion (Section 5.3), that topography is the control on climate and glaciology, then why were the two latter parameter types of data even leveraged?

If the controls on lake growth are retained, then, in the very least, only one of the factors that are dependent with another should be used in the analysis. My statistical expertise is not comprehensive (in fact, it is quite limited), but I highly recommend exploring methods such as linear mixed models or PCA. I am highly impressed by the amount of work and data presented here. From my prospective, it would really be a pity if the authors did not thoroughly explore the methods needed to make their conclusions clear, concise and well-supported.

Additionally, I may have missed it, but it is not entirely clear to me how the lakes we selected. Please explain be sure this is clearly included.

The paper is generally well-written and organized. However, at many times I found that some terms imprecise and that some reorganization is needed to make the paper more concise. The number and category of ice-marginal/ice-dammed/proglacial lakes should be more clearly explained and simplified. Also, I found much material could fit better in other sections and quite a bit of streamlining will make the paper clearer.

Hopefully my comments assist the authors in presenting this data and the findings therein. With well-executed and justified analysis and clear presentation, I envision this being a highly valuable and important paper to the community. This paper could well be useful to all sorts of researchers from glaciologists to aquatic biologists examining changes to our Earth systems as climate warms.

I am quite excited to see (and maybe cite/use) the end result of this project and wish the authors the best in their finishing this work.

**Specific Comments**

- **Title:** I would suggest a different title given the issues I mention the letter.

- **Line 13 Abstract:** *Recent work...understood* This sentence makes it sound like it is a one way process between glacier wastage and lake evolution. Might one argue it is a two-way processes with feedbacks?

- **Line 16 Abstract:** *n=107* this seems in the wrong spot. Or maybe *lake* should be plural.

- **Introduction:** I believe important information is discussed here and I found the background information adequate. However, I think some reorganization is needed. A definition of *proglacial lakes* is given in the 5th paragraph and the first paragraph, for instance. Specific terms of *proglacial, ice-marginal, ice-dammed* are discussed here but more clearly defined (in my opinion) in the Methods (Section 3). Unless I am mistaken, I also recommend that the term *proglacial* be explicitly defined as lake with glacier in contact with the water. For instance, does a lake with a proglacial area separating the lake from the glacier count? Also in the 5th paragraph *lake* is repeatedly discussed, without declaring the type.

- **Line 40** It is a matter of style, however, I find that such comments about the knowledge gap usually fit better at the end of the introduction, once the knowledge has been presented. Food for thought.

- **Line 99** *Shifting climate ... change.* It is not immediately evident where this evidence comes from, also how does this comment reconcile with the comment in line 40 about the lack of knowledge.

- **Line 104–109** I found this paragraph a little bit strange. A model of physical controls is discussed, but none is referenced. From some prospectives, a model might be presented in this work. However, I think it is more compelling to present these as "findings", as opposed to a truly generalizable model (i.e. could the code/technique/method/concept be slightly modified and applied somewhere else in the world). Also, I am concerned about the differences between physically modeling a process and statistically representing it.

- **Lines 110–115** A personal issue, which the authors may disagree with and wish to ignore. I have problems when questions begin with "how". In my opinion, it is imprecise, abstract and overly academic. Instead, I find testable questions much more interesting? "Are proglacial lakes increasing or decreasing in size? and What processes may cause variations in lake growth?" "we hope our findings will yield insights into the interactions between glaciers and downstream fluvial systems as climate warms?"

- **Line 122** One thing, which I may have missed, is how were the 107 lake selected? Here, the study area is discussed, so the number of lakes can be omitted, in my opinion. However, I found this vague in other parts of the paper.

- **Line 142–144** *Control variables, environmental parameters and predictor variables.* The way this reads, it seems like these are three terms for the same thing. Also, is any "prediction" done in the paper? I am not sure this a proper term to use here.

- **Table 1 and Sect. 2.3** It seems like *parameter* and *variable* are used somewhat interchangeably here. A parameter is a static quantity in a model, while a variable is an evolving one. I am not sure exactly how these definitions fit in to your usage later, however, please fix this and make the terms consistent. In other parts of the paper, I noticed the term *factor*. This relates to the comment above as well.

- **Line 180** *Glaciologic parameters* Same comment as above. I found some of the information here a bit beyond what is necessary for the purposes herein. It gives me confidence in your work and rigor that these things are discussed, at the same time the paper would be somewhat more concise if certain bits were omitted. For instance, do uncertainties in the GloGEM data affect your results? If so, is it best to discuss the uncertainty here, or later in the discussion when a reader may understand the interaction between your results and the GloGEM data. I would recommend stream-lining.

- **Line 202** To me, glacier response time is analysis that you conducted. Thus, if probably fits better in the Methods (Sect. 3).

- **Section 2.5** I think much of this section describes work conducted by the authors. Thus, I recommend it be transferred to the Methods (Sect. 3).

- **Line 223** This would be a result, the way it is phrased.

- **Line 233-235** This is also a method. I personally find it hard when authors discuss alternatives to their approaches, as it can make the methods hard to follow. I consider methods to be a description of what was done, and not so much a justification compared to alternatives. If you believe that your results could change substantially because of these metrics, it might be worth discussing in the context of the results in the discussion. Also, the need for an alternative method could be discussed in the introduction.

- **Line 250** Definitions of lake types are given. I think this is needed early in the paper. Also, it seems like two types of lakes exist with three definitions. *Ice-dammed and proglacial...* Then all lakes together. Are the processes so closely related that is it worth while examining the two type together (*Ice-marginal*)?

- **Line 270–272** *Due to... behavior.* This question starts to hint at how the lakes were selected. It seems like if not all lakes could be sampled that some pretty inherent biases could be in place. At one point I got the impression that these 107 lakes were all of the ice marginal lakes in the region, but this sentence suggests otherwise.

- **Section 3.2** I found that much of this section could fit in to the results section. The different characteristics of lake evolution, seem like an interesting result.

- **Section 3.3** Something of a matter of personal discretion, however, I do not think that this much information about the choice in non-parametric tests is needed. Also, for instance, I think that simply reporting the alpha value in the text will due, no need to mention here.

- **Section 4.1** I think a lot of this data could be presented nicely in a table.

- **Line 346** This again refers to my uncertainty of how the lakes were selected for study.

- **Line 358** *In term of lake number* ... number of lakes?

- **Line 399** Isn't $\tau$ already to describe glacier response time?

- **Section 4.2.1 and Table 2** I would recommend describing what summer temperature and winter precip. represent. It seems like also, water input to lakes might be an important parameter. Why is not total annual precipitation discussed? and why only summer temperature? Also, I mention this in the cover letter, but a lot of these parameters are correlated. While this is interesting, I am concerned that concluding about processes or drivers from this information is difficult. Elevation and

temperature are surely correlated. I recommend some substantially different methods to evaluate these relationships. Also, maybe it is mentioned, but what is the relationship between relative lake area change and absolute lake area change?

- **Line 498– 499** I understand the correlation here. Maybe you will get to this. However, it seems like there might be aspects of maritime topography and morphology that lend to large lake formation compared to interior areas. I hope that this will be discussed latter in the paper.

- **Section 5.1** I believe that this section would be strengthen if potential regional drivers/differences change cause variations between regions. For instance, the comparison with Wolfe makes sense because of a trend of warming light of your work, given that work goes until 2000. Also what are the differences, physically, that may cause variations between your findings and the Himalayas and Andes.

- **Lines 544** *geometric parameters... factors*? Is this topographic parameters? is this "glaciological processes?"

- **Lines 547– 563** This makes sense. However, does other work validate these findings? For instance, I assume there are papers about lake area vs. catchment area/morphology. Also, does greater glacier width increase the surface area over with frontal ablation can occur, thus creating a glacial lake faster?

- **Lines 565** Is there other work on this? Also what is the greater implication of this finding? Are estuary ecosystems changing?

- **Line 570–580** Something of a description of the landscape evolution is given, yet no papers have been cited no data or analysis provided to this end. As a result, this text must be omitted and cannot be used to support findings.

- **Line 585–589** I think this is an important topic, and I am really glad the authors are bringing it up. I hope they discuss the implications more, given paper such as Farinotti 2020, which discuss the growth of hydropower reservoirs following glacier retreat. This also has implications for GlOFs in other parts of the world.

- **Section 5.3** This section seems a bit problematic to me. Only a limited number of climatic variables were examined and the relationship between climate and glaciology is very non-linear (degree day model in the most basic sense). Does the winter precip account for more winter precip falling as rain? This is discussed in the later part of the section, but leads me to wonder why the issue was brought up in the first part of section. I suppose one motivation may be to discuss the role of topography, as opposed to climatology or glacier dynamics. However, lumping these three categories together presents something of a "chicken or the egg" problem. I recommend reconsidering this section. I discuss these issues in the cover letter.

- **Line 605** *backward climatic correlations...* inverse? also the possibility for these relationships are discussed, but no confidence interval/or correlation statistic is given. This is problematic.

- **Line 615–626** Given the non-linear reaction of glacier dynamics to climate and the justification here, I am curious why climatic parameters were explored. It seems rather post-hoc to explain why climate matters little given the correlation is small. To me, this should have been accounted for when designing the experiment.

- **Lines 643** Be careful about GLOFs. These can also occur on moraine dammed lakes and while it is beyond my expertise, these dynamics could well evolve with changing proglacial lakes.

- **Section 5.5** I think these section may need restructuring. I believe much of its content is in some way discussed above or deals with the inherent limitations or advantages of physical vs statistical modeling.

- **Lines 663** Doesn't this sentence run counter to many of the arguments presented in lines 615–626?

- **Section 6** I often consider "Conclusions" the best opportunity to position the research in the existing knowledge and state the knowledge gaps that have been filled. As a result, I am skeptical of the fact that no citations or references exist in this section.

**Figures**

- **Figure 1** I noticed this on lots of the figures, here especially. The lake area change is the close to the color of the glaciers. Can different colors be selected? also it seems a bit curious to me why lake area change, a result, is being presented here. I understand the desire to save space, but would another metric (lake area?) be better? "Detached lake" ... this seems like another term that should be defined together with the rest.

- **Figure 2** Please consider the colors again. Also, I understand the appeal of including this information. However, I am not entirely sure that I took away important findings from the figures and trends were hard to visualize given the layout. The authors could omit the figure if they desire.

- **Figure 3** I like this figure. It demonstrates the important things. Would it be worth making a couple more panels (or a cartoon) with each type of lake? Proglacial, ice-dammed, detached...

- **Figure 4** Given the choice of having the three categories of lakes above (Section 3.1, I think). Would it make sense to add a third regression line with all lakes? May the divergent behavior of the two types of lakes here suggest that the "Ice-marginal" type of lake be omitted from analysis?

- **Figure 7** I would recommend presenting this information in 2 plots. One with percip/area change and one with temp./area change. To me this plot describes more the change in climate as opposed to the effect on lakes, which is hard to see amongst the different colors and shapes.

- **Figure 8** Again, this is a somewhat difficult figure to read, and in my opinion somewhat deviates from the point of the paper, which is about lakes, not necessarily climate. Discerning a trend from the color bars is quite difficult for me, and the other information is quite intuitive and presented Figure 2. If the authors decide that this figure must stay, I recommend changing the c-axis and y-axis for the panels.

- **Figure 9** *Proglacial new* this seems like new term, possibly mentioned before. I think I understand what it is, but I recommend creating some kind of glossary early in the paper to make these things clear. Also the exclusion of "Ice-marginal" makes me think that there are two categories of lake, not 3.

- **Figure 9–11** Isn't this information a visual representation of the findings in Table 2? I think for the point of brevity and such that one or the other should be included. The other could fit well in a supplement.

---

## Author Comment (AC1)

**Reviewer 1 (Dr Jenna Sutherland)**

**General Comments**

The effects of ice-marginal lakes are a topical and emerging area of research. Due to the impacts on biophysical systems through water and sediment fluxes, the authors make a strong case for investigating physical controls on lake areal change. This is a nicely designed and detailed study which quantifies ice-marginal (both proglacial and ice- dammed) lake area changes across a representative sample over northwest North America from 1984-2018. The authors find that proglacial lakes increased in area over the last three decades whilst a large proportion of ice-dammed lakes shrunk. Specifically, large, low-elevation coastal proglacial lakes associated with wide, thick glaciers appeared to change most in absolute terms, in contrast to small, interior lakes at higher elevation that changed most relative to their initial area. Appropriate statistical analyses suggest limited correlation between climate and lake area change. Instead, the authors conclude that lake geometry and topographic setting are the dominant controls on lake area change in this region. Overall, the manuscript is well-written and clear. I suggest several very minor corrections which I expect can be addressed very easily. Although these technical corrections may look extensive, they are merely intended to help tighten up the precision of the text. I am convinced that this work presents interesting and novel results demonstrating the influence of geometry and topography as controls on ice-marginal lake evolution. Indeed, on the basis of this study I see the value of further investigations to see if similar correlations exist in other regions.

We thank Dr Sutherland for taking the time to provide a detailed review of our manuscript and address each of her specific comments below.

**Specific Comments**

I appreciate that lake area change was the focus of this study and I suspect that lake depth or bathymetry is unknown for many of the lakes. However, I wonder if the authors might comment on how important lake volume might be, relative to area, for a more in- depth discussion. A basin may well get deeper as it widens, and so lake volume is also likely to be influenced by the same environmental variables, specifically geometry and topography.

This is a great suggestion, and we have wondered in the past how this story might differ if we could measure volume change rather than area change. However, due to limited field data, with our dataset we would have to estimate volume change using an empirical relationship with area change (e.g., Cook & Quincey, 2015 - https://doi.org/10.5194/esurf-3-559-2015). Using such a relationship, volume change would scale with area change, so the outcomes should be similar, but applying this empirical relationship will make our fundamental dataset more uncertain. For the present study, we believe it is better to stick with the data we know well, rather than transforming them to be less certain if it will not fundamentally change the analysis. We have added text explaining this decision at the end of the first paragraph in Sec 5.1, copied below:

"We note that, while we here document lake area change, i.e. a readily observable quantity, similar results would likely emerge if we estimated lake volume change because lake area scales with lake volume (Cook and Quincey, 2015; Shugar et al., 2020). However, converting area to volume in the absence of field observations requires the use of empirical scaling relationships (Cook and Quincey, 2015; Shugar et al., 2020) and would make the presented data more uncertain; we therefore only consider area change in the present study."

The statistical analyses undertaken and assumptions made are valid and clearly outlined. However, the authors infer that covariance between continentality and climatic parameters is the underlying cause for correlation with other environmental factors. Line 660 states that 'when considering climatological, glaciological and topographic controls on lake area change, it is important to note that these variables are often intertwined'. I wonder why a multivariate statistical technical such as Principal Component Analysis was not undertaken instead? In PCA, the data are easily reduced into smaller numbers of interrelated groups that can reveal underlying patterns within the dataset. Measuring the importance of each variable relative to each other in this way might help to confirm the authors assumptions.

Thank you for this suggestion, which echoes one of the main comments from Reviewer 2. In the revised manuscript, we incorporate principal components analysis and a more thorough investigation of covariance between the environmental variables. These new analyses resulted two entirely new sections, one new figure and table in the main text, as well as one new figure and table in the supplement. The revised manuscript includes two entirely new aspects to address thhs comment.
(*The following text is identical to that adressing the similar comment from Reviewer 2*).
We now include both: 1) in-depth discussion of covariance between environmental variables, and 2) an entirely new analysis undertaking principal components (PC) analysis to reduce data dimensionality and then running correlations against the PC scores. The discussion of environmental variable covariance highlights the difficulty in untangling some of these variables, as noted in the first paragraph. We believe these data, as well the new figure & existing table associated with them (that is now more centrally discussed), provide stronger support for our existing claims, but also better highlight uncertainty in determining causality between a single environmental variable and ice-marginal lake area change. The PCA results are consistent with our bivariate results, and provide stronger support for the dichotomous behavior of large, coastal, low-elevation lakes and small, interior, high-elevation lakes. The PCA results also help to disentangle the relationship between glacier attributes (e.g., glacier area, lake-adjacent ice thickness) and topographic attributes (e.g., distance from coast, elevation). Topographic variables load strongly onto the second principal component axis (PC2,) with minimal influence from glaciologic variables, while the opposite is true for PC3 (Table S2 in the revised manuscript). The significant correlation between lake area change metrics and PC2 scores (which holistically reflect

continentality), and very limited significant associations with PC3 scores (which reflects glacier size), supports the notion that topography is more closely associated with lake area change than glaciologic characteristics. We have updated the text to clearly state these observations. The PCA data are somewhat more abstract than the bivariate analyses, though, so we believe it important to present both bivariate and multivariate analyses to provide more compelling and physically-meaningful evidence of our claims than either analysis would provide alone.

The new sections, figures, and tables devoted to these analyses are copied below, with more support in the supplemental material and in text throughout the manuscript:

[revised manuscript text omitted]

Technical corrections

Line 23. 'These systems...' Could you clarify which systems are you referring to here. The low-elevation coastal proglacial lakes or interior lakes at high elevation? Or both?

We clarified this statement by saying "the fastest changing lakes" rather than "these systems".

L28. The Introduction is labelled as 1.1 but there are no subheadings within this section. Change to 1.

We removed the subsection label.

L37. 'may pose a serious hazard'. To downstream communities?

Added.

L56-60. Make it clear that this study (Wolf et al. 2014) refers to lakes in northwest North America (I think)

We added a clarification that the Wolf et al. (2014) study focuses on Alaska.

L62-63. This sentence seems to be a repeat of what is stated in the first sentence of the intro (lines 29-30). I think you can delete either one to make the text more concise.

We modified this to transition sentence to avoid repetition with the earlier text that has been highlighted. It now reads: "The development and evolution of an ice-marginal lake can impact its associated glacier".

L70. 'bys' should be 'by'

Thanks for your careful read. We corrected this typo.

L75. 'In addition to these glaciological factors....' It seems unusual to start a new paragraph like this. This sentence should be linked to the points made above.

We removed this opening clause and instead jump directly into ecosystem impacts.

L79. 'below lakes.' I suggest rephrasing this because it reads as if the streams are positioned underneath the lake, I think you mean downstream?

Changed "below lakes" to "downstream from lakes".

L83-86. A bit of repetition (such GLOFs) from paragraph 1 here. I suggest paragraph 4 of the intro is either deleted or at least condensed and moved into another paragraph. I would also suggest combing the last two (short) paragraphs of the intro into one.

We removed the unnecessary GLOF text from the fourth paragraph. We also combined the last two paragraphs, as suggested. However, we choose to leave the paragraph discussing geomorphic and ecological impacts of ice-marginal lakes as its own paragraph. These impacts are very distinct from glaciological impacts (the previous paragraph) and, despite their importance, often receive minimal discussion in studies of ice-marginal lakes. We believe it is important to highlight their role in the "icefield-to-ocean" biophysical system.

L94. 'how lakes change over time'. And space?

Added.

L98. I suggest clarifying here if the lakes studied in Brun et al. (2020) were ice-marginal or disconnected/distal to glacier terminus. This is quite important. I think the latter.

Correct - we added a qualifier saying that Brun et al. (2020) was focused on closed basin lakes that are not in direct contact with glacier ice.

L100. 'shifted up in elevation'. I suggest rephrasing for clarity, e.g. shifted to higher elevations.

Done.

L101. 'may influence proglacial lake evolution...' add 'also' in-between may and influence

Done.

L106. Clarify if this 'model' is a numerical/physical/conceptual model?

Clarified that we mean a conceptual model, though a numerical model would be even better!

L122. 'Our 107 study lakes span...' I suggest re-phrasing, so it reads more fluently e.g. 'We study 107 lakes that span...'

Done.

L163. 'We utilize the 1960s decade to consider the longest-term comparison allowed by the SNAP dataset.' Given that the lake extents were digitised from 1980s onwards, could you have utilised 1980s decade also?

Our thinking here is that, because glaciers don't respond instantaneously to climate change (echoing discussion in Sec 5.3, 2nd paragraph), that using the longest-term climate data give us the best chance of describing climate change on the timescale relevant to glacier change. We note that, due to data limitations, our glacier mass balance model only extends back to 1980 (rather than 1960) and so the cumulative mass balance dataset we correlate with is, in fact, computed over the same timescale as our area change data.

L267. 'summery' should be 'summer'

Done.

L314. 'We exclude lakes that detached from their adjacent glacier (n = 18; 13 proglacial lakes and 5 ice-dammed lakes).' These numbers don't seem to match up with what is written in lines 257-259 that suggests 15 proglacial lakes detached?

We have worked through the entire manuscript to ensure consistency of lake numbers. We apologize for this error that originated from slight changes as we iterated on earlier versions of this dataset and manuscript.

L318. 'Though we omit these lakes from lake area change characterization and analysis'. I would delete this as it is repeated from the previous sentence and perhaps re-phrase this part of the paragraph, so it reads more fluently.

We deleted that redundant clause.

L358. I found the many values reported in the first two paragraphs of section 4.1 slightly confusing as to what they represented. E.g. Lines 358 'In terms of lake number, 83 % of the investigated proglacial lakes (n = 88 in total) grew, 10 % shrunk, and 7 % were relatively stable, changing by less than ± 0.1 km2.' So what then are the numbers reported for proglacial lakes in previous paragraph (Lines 349-350)? Are the 72 proglacial lakes that grew, 82 %? Do these statements need to be linked in some way?

As with your comment to L314, these issues stem from slight changes during data processing & analysis on an earlier version of the manuscript & dataset that we neglected to update for the submission. We apologize for this error and have combed through the manuscript to ensure all numbers are consistent and derive from the most recent data processing.

L493. 'Though their signs are flipped'. Could there be a better way to phrase this?

We now say "with opposite signs".

L518. 'against the backdrop...' I suggest re-phrasing this e.g. 'into the context of global...' or similar

Done.

L554-556. 'The initial existence of a large lake requires a large basin, and basins generally do not end abruptly. Therefore, the simple existence of a large lake suggests that there is high potential growth in a regionally-extensive depression.' I agree with this statement to a certain extent, but the opposite could also be true – initially large lakes could be at their maximum extent with no further room for growth and therefore I would not expect large lakes to expand further.

We added a clarification that this first interpretation requires proglacial lakes to be in an early stage of development, and point to the relevant study by Emmer et al. (2020) which elaborates on this

idea of stages of lake development. You are correct that this nuance is required for this hypothesized mechanism to be better supported. The text now reads:

"1) The initial existence of a large lake requires a large basin, and basins generally do not end abruptly. Therefore, the simple existence of a large lake suggests that there is higher potential growth in a regionally-extensive depression. This explanation would require Alaska's proglacial lakes to be in an early stage of development (Emmer et al., 2020), with ample room to grow into overdeepened basins."

L556. 'longer zones' – clarify whether this is spatially or temporally. I think you mean how much surface area of the glacier is in contact with the lake as 'wider' is referred to later in the sentence.

We changed this hypothesis to now read "Alternatively, 2) larger lakes likely have greater surface area at the glacier-lake interface, which may lead to higher rates of frontal ablation".

L559. 'perhaps that large lakes tend to be warmer'. Could a reference be cited here?

Two references have been added to support this hypothesis that larger lakes tend to be warmer, both using the example of large Patagonian proglacial lakes.

L560. 'submarine' change this to one of subaqueous/basal/sub-lacustrine (my preference is subaqueous)

We changed "submarine" to "subaqueous" throughout the manuscript. Thank you for noting this lack of precision.

You infer that topographic and geometric factors most strongly control ice-marginal lake area change, so perhaps in section 5.2 a discussion about how the geometry of the lake and the topography also influence the geometry of the adjacent glacier could be made. The following study supports your conclusions about such inter-related variables:

Sutherland, J. L., Carrivick, J. L., Gandy, N., Shulmeister, J., Quincey, D. J., & Cornford, S. L. (2020). Proglacial lakes control glacier geometry and behavior during recession. Geophysical Research Letters, 47(19), e2020GL088865.

Thank you for pointing us to this very interesting recent paper of yours - we were not previously aware of it. We agree it is very relevant for the discussion we present here and have added references to it in Sections 1. We do not add a reference in Sec 5.2 because our impression is that this study more shows the impact proglacial lakes have on glacier geometry relative to land-terminating glaciers (rather than compared to other proglacial lakes with differing basin geometry).

We looking through the main text and supplements (including sensitivity analyses) of this paper and did not find mention of the impact of basin geometry on glacier dynamics, though we can certainly imagine the concept you're referring to.

L556. 'Several other factors are statistically significantly linked can be explained using...' Insert 'that' after factors.

Done (this was L565 in the original manuscript, not 556 as written).

L621. 'glaciers act as low-pass filters on climate variability'. Could you elaborate upon this? Do you mean that subtle changes in climate are not detected by the glacier?

We deleted this statement, which added unnecessary confusion and is not essential.

L623. 'responding to climate change in a lagged and smoothed manner' Insert 'is' before 'responding'

Done.

L637. 'lake area coming into equilibrium with the current environment' I am unsure what you mean here, equilibrium suggests a steady state so do you mean the lake will neither grow or shrink further?

We clarified that we do mean a steady state. We also removed this subsection, moving relevant text to Sec 5.1. We also moved the figure supporting these claims to the supplementary material because it is not essential to the main drive of our story and we strove to streamline the manuscript as much as possible.

L669. 'where has been more winter precipitation' should be 'where there has been...'

Done.

L698. There is a change of tenses within this sentence. I suggest changing 'find the majority (82 %) of proglacial lakes are growing' into the past tense.

Done.

Code data and availability.

I am unable to access the data specifically linked to these webpages : https://arcticdata.io/xx and https://github.com/armstrwa/xx (the end of the links appear to be missing?).

We have corrected these placeholder links with active links.

'Climate reanalysis data are available at xx'. Insert the correct location instead of xx?

Thank you for catching this. We replaced "xx" with the link that was incorrectly listed for the RGI (as you note below).

'The Randolf Glacier inventory is located at [link]' this link actually takes you to the SNAP climate data not RGI.

We changed this to the correct link. Thank you for noting this issue.

Figures and Tables

Figure 1. Perhaps an insert of the wider region would be useful here for readers not familiar with the area? Could you state what the insert (photograph) is in the caption, e.g. is this glacier or lake named?

We substantially revised this figure. It now presents lake locations by their type (i.e., proglacial or ice-dammed) rather than results (i.e., lake area change), as originally presented, which addressed a comment by Reviewer 2. In the revised figure we include an example of both a proglacial and ice-dammed lake, and provide their locations in the caption. The revised figure (copied below) also shows a map inset to give broader geographic context.

[Figure]

Figure 3. Lines 242. 'Ice thickness color bar and map scale are identical between panels a and b'. I don't think you need to state this in the caption because the reader can see they are identical looking at the figure. If space allows, could the lake outlines from different years also be labelled (I think I see 3 or 4 different outlines)?

We removed the text about the colorbar and scale. We did not add year labels to the lake outlines at each timestep to avoid making an already busy figure more complicated, but the revised Figure 2 now includes examples of proglacial and ice-dammed lake area change with chronologically labelled lake extents. We hope this provides the kind of example you are asking for in this comment.

Figure 4. Line 373 in the caption 'while proglacial lakes that appeared that time are unfilled' is there something missing here? The dashed line shows 1:1 (i.e., lakes with constant area), while the dashed lines show various levels of relative area change.' Do you mean dotted lines perhaps?

We clarified this caption to now read "Proglacial lakes that existed for the entire study period are shown as filled blue circles, while proglacial lakes that appeared that time ("new lakes") are shown as unfilled symbols" and fixed the noted dashed/dotted typo.

Figure 6a. Has this data been plotted from specific lakes? If so, could the lake/glacier be labelled or mentioned in the caption?

We have moved this figure to the supplements because it was not essential for conveying our main findings.

Table 1. Could you add the resolution (and range) of elevation data from GTOPO30 here?

GTOPO30 has 30 arc-second (~1 km) spatial resolution and global extent. We specify these values in Sec 2.5, but believe adding this information to Table 1 would be strange because none of the other data have their resolution stated here. Statements of resolution, extent, and accuracy are made in Secs 2.2-2.5, and incorporating this information into Table 1 would be difficult because: 1) it would make the table very dense, and 2) resolution of some datasets does not have a fixed spatial resolution because they are calculated on a glacier-wide basis.

Supplementary Information

Line 928. Figure S1 caption, insert the word 'lake' in-between 'proglacial bounded...'

Added.

Figure S7. I can't see where this figure is referred to in the main text. Figure caption states c) is 'winter precipitation for each lake for 2009-2009 decade'. Should this be 2000-2009 instead?

We added a reference to this figure in Section 5.3 while discussing correlations between relative area change and climate parameters.

Line 984. 'A positive change indicates warmer temperatures' I think needs to be deleted as it is repetition of c) and d) is precipitation only

We changed this to state that a positive value in panel d indicates wetter winters in recent times.

Line 990. 'lake are time series and area change' should be 'lake area time series'?

Thank you, yes. We changed "are" to "area".

Dr Jenna Sutherland (Leeds Beckett University, UK). 05/02/2021.

---

## Author Comment (AC2)

**Reviewer 2**

We thank this anonymous reviewer for their careful read and constructive comments and suggestions.

**General Comments**

Dear Editor,

Thank you for the opportunity to review this paper. The authors leverage a record of Landsat imagery to examine the evolution of ice-marginal lakes over the last 3 decades for the Gulf of Alaska. Changes to lake size are compared with available data about glacier dynamics, melt, glacier morphology, topographic conditions and climate data to better understand possible drivers of changes to glacier lake size.

The paper covers important topics that must be approached and I commend the authors for pursuing this original research. Clearly a large amount of novel data was ascertained and analyzed in the context of existing datasets. However, I found some substantial issues with the text which either require clarification or methods need to be adjusted.

In my opinion, the methods and conclusions are generally well-explained. However, I am skeptical of the conclusion that topography is the primary control on ice marginal lake change, given that the authors point to the interaction of topography on climate and glaciological parameters. This seems like a bit of circular argument. Successfully arguing this case will require evaluating the interaction between climate, topography and glaciology and the result deemed independent (or not). If, as stated in the discussion (Section 5.3), that topography is the control on climate and glaciology, then why were the two latter parameter types of data even leveraged?

If the controls on lake growth are retained, then, in the very least, only one of the factors that are dependent with another should be used in the analysis.

My statistical expertise is not comprehensive (in fact, it is quite limited), but I highly recommend exploring methods such as linear mixed models or PCA. I am highly impressed by the amount of work and data presented here. From my prospective, it would really be a pity if the authors did not thoroughly explore the methods needed to make their conclusions clear, concise and well-supported.

Thank you for the two suggestions, which are closely related and also echo one of the main comments from Reviewer 1. The revised manuscript includes two entirely new aspects to address these concerns.
*(The following text is identical to that addressing the similar comment from Reviewer 1).*
We now include both: 1) in-depth discussion of covariance between environmental variables, and 2) an entirely new analysis undertaking principal components (PC) analysis to reduce data

dimensionality and then running correlations against the PC scores. The discussion of environmental variable covariance highlights the difficulty in untangling some of these variables, as you note in the first paragraph. We believe these data, as well the new figure & existing table associated with them (that is now more centrally discussed), provide stronger support for our existing claims, but also better highlight uncertainty in determining causality between a single environmental variable and ice-marginal lake area change. The PCA results are consistent with our bivariate results, and provide stronger support for the dichotomous behavior of large, coastal, low elevation  lakes and small, interior, high elevation lakes. The PCA results also help to disentangle the relationship between glacier attributes (e.g., glacier area, lake-adjacent ice thickness) and topographic attributes (e.g., distance from coast, elevation). Topographic variables load strongly onto PC2, with minimal influence from glaciologic variables, while the opposite is true for PC3 (Table S2 in the revised manuscript). The significant correlation between lake area change metrics and PC2 scores (which holistically reflect continentality), and very limited significant associations with PC3 scores (which reflects glacier size), supports the notion that topography is more closely associated with lake area change than glaciologic characteristics. We have updated the text to clearly state these observations. The PCA data are somewhat more abstract than the bivariate analyses, though, so we believe it important to present both bivariate and multivariate analyses to provide more compelling and physically-meaningful evidence of our claims than either analysis would provide alone.

*We omit pasting the new sections, figures, and tables associated with these analyses, which can be found in our response to Reviewer 1.*

Additionally, I may have missed it, but it is not entirely clear to me how the lakes we selected. Please explain be sure this is clearly included.

Thank you for noting that this was not clear. We choose a random sampling of ice-marginal lakes using zoomed-in Landsat 8 imagery. We strove to avoid spatial clustering of lakes by using a gridded map and including a near-uniform amount of lakes within each grid cell. We strove to sample from a range of lake sizes, and it is evident that we succeeded in this from analysis of our lake area histograms shown in Figure 4. We now present these study site selection methods early in Sec 3.1 to make these methods clearer. The first paragraph of Sec 3.1 now includes text stating:

"We utilize a gridded map and select a similar number of lakes in each grid cell to avoid biased site selection and clustering. A subset of lakes (n = 40) is sampled from a historical catalog of ice-marginal lakes in Alaska (Post & Mayo, 1971) to avoid undersampling lakes that disappeared and could not be observed in recent satellite imagery."

The paper is generally well-written and organized. However, at many times I found that some terms imprecise and that some reorganization is needed to make the paper more concise. The number and category of ice-marginal/ice-dammed/proglacial lakes should be more clearly explained and simplified. Also, I found much material could fit better in other sections and quite a bit of streamlining will make the paper clearer.

We have reorganized the text, particularly in the results & discussion sections. In this revision, we sought to cut redundant language, instead focusing it in single sections as much as possible. We include a new summary table at your suggestion (described below) which further helps simplify presentation of these data.

Hopefully my comments assist the authors in presenting this data and the findings therein. With well-executed and justified analysis and clear presentation, I envision this being a highly valuable and important paper to the community. This paper could well be useful to all sorts of researchers from glaciologists to aquatic biologists examining changes to our Earth systems as climate warms.

I am quite excited to see (and maybe cite/use) the end result of this project and wish the authors the best in their finishing this work.

We are grateful for the encouragement and would like to thank the reviewer again for these thorough comments.

**Specific Comments**
Title: I would suggest a different title given the issues I mention the letter.

We have changed the manuscript's title to "Gulf of Alaska ice-marginal lake area change over the Landsat record and potential physical controls" to avoid over-selling our certainty in topographic controls.

Line 13 Abstract: Recent work...understood This sentence makes it sound like it is a one way process between glacier wastage and lake evolution. Might one argue it is a two-way processes with feedbacks?

This sentence now reads "Recent work suggests positive feedbacks between glacier wastage and ice-marginal lake evolution."

Line 16 Abstract: n=107 this seems in the wrong spot. Or maybe lake should be plural.

We moved n = 107 to after "lake perimeters" to avoid singular-plural confusion.

Introduction: I believe important information is discussed here and I found the background information adequate. However, I think some reorganization is needed. A definition of proglacial lakes is given in the 5th paragraph and the first paragraph, for instance. Specific terms of proglacial, ice-marginal, ice-dammed are discussed here but more clearly defined (in my opinion) in the Methods (Section 3). Unless I am mistaken, I also recommend that the term proglacial be explicitly defined as lake with glacier in contact with the water. For instance, does a lake with a proglacial area separating the lake from the glacier count? Also in the 5th paragraph lake is repeatedly discussed, without declaring the type.

We removed the redundant definition of proglacial lakes at the beginning of what was the 5th paragraph in the original submission. We also reorganized the introduction by moving this paragraph up (now appearing as the 4th paragraph) to keep glacier-related items together. The introduction now flows more logically, with the 3rd paragraph discussing impacts of lake change on glaciers and the 4th paragraph discussing impacts of glacier change on lakes. In the new transition sentence to the 4th paragraph, we now stress that glacier-lake interactions are a two-way process, as you mentioned in your comment about the abstract. We note that in the track changes document much of this text shows up a "new" rather than "moved" due to slight changes in wording in places.

Line 40 It is a matter of style, however, I find that such comments about the knowledge gap usually fit better at the end of the introduction, once the knowledge has been presented. Food for thought.

Thanks for this suggestion. We believe it is important to keep this knowledge gap statement in the first paragraph to motivate our work as quickly as possible, rather than leave the reader waiting to see what it is we do. At the end of the introduction, we then pose the specific research questions stemming from the broader statement in the first paragraph. We choose to retain this structure because we believe it engages the reader early, refocuses them later, and balances broadness and specificity.

Line 99 Shifting climate ...change. It is not immediately evident where this evidence comes from, also how does this comment reconcile with the comment in line 40 about the lack of knowledge.

We simplified this sentence, removing the "shifting climate conditions" part, and merged it with the following sentence. That sentence's function was simply to transition to Alaska-based studies, and the modified version does this more clearly.

Line 104–109 I found this paragraph a little bit strange. A model of physical controls is discussed, but none is referenced. From some prospectives, a model might be presented in this work. However, I think it is more compelling to present these as "findings", as opposed to a truly

generalizable model (i.e. could the code/technique/method/concept be slightly modified and applied somewhere else in the world). Also, I am concerned about the differences between physically modeling a process and statistically representing it.

Thanks for pointing out this lack of clarity. The other reviewer also keyed into this statement. In the revised text, we clarify here that we mean a conceptual model rather than a specific numerical model. We very much view our work as the first steps in a larger effort that could eventually culminate in a physics-based numerical model, but these observations and analyses are just a starting point.

Lines 110–115 A personal issue, which the authors may disagree with and wish to ignore. I have problems when questions begin with "how". In my opinion, it is imprecise, abstract and overly academic. Instead, I find testable questions much more interesting? "Are proglacial lakes increasing or decreasing in size? and What processes may cause variations in lake growth?" "we hope our findings will yield insights into the interactions between glaciers and downstream fluvial systems as climate warms?"

We do choose to leave the "how" question as is, but I understand the reviewer's perspective. We wish to cast these questions broadly, but we do also state more specific/testable questions following the broad ones in that paragraph.

Line 122 One thing, which I may have missed, is how were the 107 lake selected? Here, the study area is discussed, so the number of lakes can be omitted, in my opinion. However, I found this vague in other parts of the paper.

We revised this statement to say we "study a sampling of ice-marginal lakes" across northwestern North America and now just include the sample size as a parenthetical. The reason that we wish to leave the sample size here is to be clear that we did not digitize every lake in the region. To avoid spatial clustering in study lakes, we used a gridded map and selected a constant number of lakes from each grid cell. The number of 107 lakes is somewhat arbitrary, but was found to balance the needs for a relatively large sample size for robust statistical analyses with the time-intensive work of digitizing lake perimeters and extracting the relevant datasets. We describe the site selection process in Section 3.1, and have now moved the text to appear earlier in that section. This revised text appears in our response to your main comment about this issue.

Line 142–144 Control variables, environmental parameters and predictor variables. The way this reads, it seems like these are three terms for the same thing. Also, is any "prediction" done in the paper? I am not sure this a proper term to use here.

Yes, you are correct that we used these terms more or less interchangeable. In the revised manuscript (and in response to one of your main comments), we avoid saying "controls" due to the complicated dependencies between environmental variables and the fact that our statistical analyses show correlation but not control/causation. We mean "predictor" in the statistical sense, in which it is essentially synonymous with "independent variable". If there is a statistical association between two quantities, the predictor (independent variable) can be used to predict the value of the dependent variable. We choose to retain the use of "predictor variable" just to avoid redundancy in saying "environmental variable" over-and-over again throughout the text.

Table 1 and Sect. 2.3 It seems like parameter and variable are used somewhat interchangeably here. A parameter is a static quantity in a model, while a variable is an evolving one. I am not sure exactly how these definitions fit in to your usage later, however, please fix this and make the terms consistent. In other parts of the paper, I noticed the term factor. This relates to the comment above as well.

Thank you for pointing out that we were not being precise. In the revised manuscript, we just call all of these quantities "variables" to avoid confusion. All of these quantities evolve in time over sufficiently long timescales.

Line 180 Glaciologic parameters Same comment as above. I found some of the information here a bit beyond what is necessary for the purposes herein. It gives me confidence in your work and rigor that these things are discussed, at the same time the paper would be somewhat more concise if certain bits were omitted. For instance, do uncertainties in the GloGEM data affect your results? If so, is it best to discuss the uncertainty here, or later in the discussion when a reader may understand the interaction between your results and the GloGEM data. I would recommend streamlining.

Thank you for seeing this language as adding rigor to the study and increasing your confidence in it. In our perspective, it is important to include these brief synopses of the datasets and their associated uncertainties because most of these products heavily rely on model data in an area with sparse ground observations. Later, the fact that we do/do not find a statistical relationship between two quantities (e.g., summer temperature change and lake area change) could be because, in fact, no relationship exists, or because our estimates of temperature change are in error. We discuss this idea in Sections 4.4 and 5.4 and believe retaining these statements here provides the reader with more information to make their own decision about the reliability of these results and datasets.

Line 202 To me, glacier response time is analysis that you conducted. Thus, if probably fits better in the Methods (Sect. 3).

The calculation of glacier response time very closely stems from Huss and Hock (2015), cited in that sentence. You are correct in that it is technically a new analysis because those results were not published in that paper, but we feel that including this short statement in the Methods would be confusing and distract from the main drive of that section. This drive of this paper is ice-marginal lakes, not glacier modeling, and we feel a separate section devoted to this relatively short statement would seem out of place.

Section 2.5 I think much of this section describes work conducted by the authors. Thus, I recommend it be transferred to the Methods (Sect. 3).

We view those analyses using topographic data as pre-processing for later statistical analyses. To reflect the fact that Section 2 largely describes datasets, but also includes some minor processing steps, we now title it "Study area, datasets, and data pre-processing". This may also partly address your comment above concerning glacier response time.

Line 223 This would be a result, the way it is phrased.

We did not mean this statement as definitive, but rather just meant to motivate why we extract ice thickness in the region immediately abutting the lake. The statement now reads "However, it is plausible that glacier attributes in the region immediately bordering an ice-marginal lake may be more important for the lake's evolution." We believe there is less likelihood that this will be interpreted as a result with the "it is plausible" statement.

Line 233-235 This is also a method. I personally find it hard when authors discuss alternatives to their approaches, as it can make the methods hard to follow. I consider methods to be a description of what was done, and not so much a justification compared to alternatives. If you believe that your results could change substantially because of these metrics, it might be worth discussing in the context of the results in the discussion. Also, the need for an alternative method could be discussed in the introduction.

We deleted the statement about alternative strategies, and agree it likely just reduces clarity/concision. Again, we hope the reframing of this section to include pre-processing steps for statistical analyses makes inclusion of these steps in this section more agreeable.

Line 250 Definitions of lake types are given. I think this is needed early in the paper. Also, it seems like two types of lakes exist with three definitions. Ice-dammed and proglacial... Then all lakes together. Are the processes so closely related that is it worth while examining the two type together (Ice-marginal )?

We include somewhat-broader parenthetical definitions of proglacial and ice-dammed lakes in the first sentences of the introduction, and here simply provide a little more detail. We do believe it is justifiable to split *a priori* proglacial and ice-dammed lakes for later analyses. The relationship between these kinds of lakes and their associated glaciers is fundamentally different, and one would reasonably expect them to respond differently to climate & glacier change. Our results show the two groups have distinctly different distributions of area change (Fig 5 in the original manuscript). Further, our statistical analyses later show several correlations where the sign of a statistically-significant correlation differs between proglacial and ice-dammed lakes (e.g., initial lake area; Table 2 in the original manuscript). Combining the two lake types would add noise that could obscure physically-meaningful trends.

Line 270–272 Due to... behavior. This question starts to hint at how the lakes were selected. It seems like if not all lakes could be sampled that some pretty inherent biases could be in place. At one point I got the impression that these 107 lakes were all of the ice marginal lakes in the region, but          this          sentence          suggests          otherwise.

It is correct that we did not digitize all lakes  in the region, and have added text to clarify that we investigate a sampling of the region's ice-marginal lakes. We discuss how we selected these sites in response to the comment about Line 122. We also state how we sought to avoid study site selection bias in Lines 272-273 of the original manuscript.

Section 3.2 I found that much of this section could fit into the results section. The different characteristics     of     lake     evolution,     seem     like     an     interesting     result.

Yes, it was a little difficult to decide where to put these statements. We have to give the reader enough context to know what we're doing and why, but ultimately, these statements are meant to motivate the methods we then describe (e.g., curve fitting). We removed the statements with specific numbers (e.g., 9 proglacial lakes formed) because, like you mention, these are results and they also appear in Section 4.1 (Lines 477-483 of the original manuscript).

Section 3.3 Something of a matter of personal discretion, however, I do not think that this much information about the choice in non-parametric tests is needed. Also, for instance, I think that simply     reporting     the     alpha     value     in     the     text     will     due,     no     need     to     mention     here.

We believe discussing the non-parametric statistics is important because they will be less familiar to many readers (than Pearson linear correlation) and the choice of test makes a big difference for final results. For example, when the tests we describe here are run through Pearson correlation, many more statistically-significant relationships appear, but that is just because outliers and non-normal distributions exist in many of these datasets, and Pearson is more sensitive to these things than Kendall.

Section 4.1 I think a lot of this data could be presented nicely in a table.

This is a great suggestion. We added such a table (Table 2 in the revised manuscript) and now include references to this table throughout the text, which allows us to cut bulky, number-laden language elsewhere. This new table is copied below.

Table 2. Summary statistics for proglacial and ice-dammed study lake area change. Steady lakes are defined as having changed by less than ± 0.1 km². Summary statistics are shown for the change of individual lakes, as well as the cumulative area of all study lakes. For descriptors of individual lakes, we use the robust statistics of the median and $10^{th}$ and $90^{th}$ percentile lake area change because the existence of extreme values makes the minimum, mean, and maximum area change less meaningful. Relative area change is scaled by a lake's initial area, so a 100% increase indicates a lake that doubled in area, while -100% indicates a lake that completely disappeared.

| | Proglacial | | | Ice-dammed | | |
|---|---|---|---|---|---|---|
| Number of lakes | Growing | Steady | Shrinking | Growing | Steady | Shrinking |
| (- , %) | 72 (83%) | 11 (13%) | 4 (5%) | 3 (15%) | 8 (40%) | 9 (45%) |
| Absolute area change | 10th % | Median | 90th % | 10th % | Median | 90th % |
| (individual, km²) | 0.01 | 1.28 | 6.76 | -3.7 | -0.04 | 0.36 |
| Relative area change | 10th % | Median | 90th % | 10th % | Median | 90th % |
| (individual, %) | 8% | 125% | >1000% | -82% | -15% | 212% |
| Cumulative area | 1984 | 2018 | Change | 1984 | 2018 | Change |
| (km²) | 336 | 606 | 270 (81%) | 96 | 80 | -17 (-17%) |

Line 346 This again refers to my uncertainty of how the lakes were selected for study.

It is our intention with this statement is exactly to point out that we do not digitize every lake in the region. We are not exactly sure, however, what/how the reviewer would like us to change this statement.

Line 358 In term of lake number . . . number of lakes?

Changed.

Line 399 Isn't τ already to describe glacier response time?

We changed the symbol for glacier response time to "T" and retain τ for the correlation strength.

Section 4.2.1 and Table 2 I would recommend describing what summer temperature and winter precip. represent. It seems like also, water input to lakes might be an important parameter. Why is not total annual precipitation discussed? and why only summer temperature? Also, I mention this in the cover letter, but a lot of these parameters are correlated. While this is interesting, I am concerned that concluding about processes or drivers from this information is difficult. Elevation and temperature are surely correlated. I recommend some substantially different methods to evaluate these relationships. Also, maybe it is mentioned, but what is the relationship between relative lake area change and absolute lake area change?

We have added clarifications in Sec 4.2.1 and Sec 5.3 that we run correlations with winter precipitation and summer temperature (and their change) because these are the climatic variables most closely associated with glacier mass balance, which presumed would be the most relevant for predicting ice-marginal lake area change, though our results show that to not be the case. In Sec 5.3, we now make the following statement.

We note that we do not run correlations between lake area change and mean annual precipitation because variations in winter precipitation and summer temperature show strong relationships with Alaska glacier mass balance, particularly for coastal glaciers (e.g,. McGrath et al., 2017), though changes in precipitation throughout the whole year could be more important for glacier mass balance elsewhere. Probing relationships with environmental variables beyond those presented here provide productive avenues for future research.

As mentioned in our response to the first major comment, the revised manuscript includes more discussion of covariance between environmental variables, clearly states where some variables are very difficult to untangle, and provides new multivariate statistical analyses to support our argument that topographic factors appear more important than glaciological or climatological factors in predicting lake area change.

Relative area change is the absolute area change divided by the lake's initial area. We describe this in the first paragraph of Section 3.2 ("Lake area change analysis"), copied below.

We determine absolute lake area change ($\Delta A$) as the simple difference in area between our last and first lake delineations, where a positive area change indicates a growing lake … We determine relative lake area change as $\Delta A / A_0$, where $A_0$ is the lake's first observed area and $\Delta A$ represents the absolute change in lake area over the study period.

Line 498– 499 I understand the correlation here. Maybe you will get to this. However, it seems like there might be aspects of maritime topography and morphology that lend to large lake formation compared to interior areas. I hope that this will be discussed latter in the paper.

Yes, we share the same thoughts and discuss them in Section 5.2. We are trying to keep this Section 4.3 (the section you refer to above) as close to pure results as possible, and so do not elaborate on that finding at that point.

Section 5.1 I believe that this section would be strengthen if potential regional drivers/differences change cause variations between regions. For instance, the comparison with Wolfe makes sense because of a trend of warming light of your work, given that work goes until 2000. Also what are the differences, physically, that may cause variations between your findings and the Himalayas and                                                                                                                      Andes.

We added discussion into how our results may be similar or differ to other areas depending on the relative stage of proglacial lake development, debris cover on glaciers, and methodological differences.

Lines 544 geometric parameters... factors? Is this topographic parameters? is this "glaciological processes?"

We changed "geometric parameters" to "topographic variables". The other reference is to glaciologic variables and not glaciological processes.

Lines 547– 563 This makes sense. However, does other work validate these findings? For instance, I assume there are papers about lake area vs. catchment area/morphology. Also, does greater glacier width increase the surface area over with frontal ablation can occur, thus creating a glacial lake faster?

We only mean to hypothesize potential explanations for the correlation between initial lake area and lake area change, not to state that this is a fact. We have modified the text to highlight that we are just taking an educated guess here, now stating "We hypothesize two mechanisms that may explain [this association]...". We added reference to existing work that discusses the idea of stages of proglacial lake evolution and a caveat that the first hypothesis we present requires that Alaska's lakes are in an early stage of development with ample room to grow into overdeepened basins. However, we do not know of any study that systematically assesses the relationship between ice-marginal lake area and potential subglacial catchment area to be able to provide context/verification for this hypothesis. In response to your second question, we would expect the contact area between glacier ice and lake water to scale with glacier width, likely nonlinearly with an exponent > 1, because wider glaciers tend to be thicker glaciers. The idea you are getting at is what we meant to convey in our second hypothesis, which we have clarified. It now reads "Alternatively, 2) larger lakes likely have greater surface area at the glacier-lake interface, which may lead to higher rates of frontal ablation".

Lines 565 Is there other work on this? Also what is the greater implication of this finding? Are estuary ecosystems changing?

We are not aware of any other working reporting statistical analyses of topographic factors linked to ice-marginal lake area change.

Line 570–580 Something of a description of the landscape evolution is given, yet no papers have been cited no data or analysis provided to this end. As a result, this text must be omitted and cannot be used to support findings.

We added references to Péwé (1975) "Quaternary geology of Alaska" and Kaufman & Manley (2004) "Pleistocene Maximum and Late Wisconsinan glacier extents across Alaska, U.S.A." to support these "zoomed-out" statements about Alaska's glacial history and the general structure of the landscape encountered across a transect from coast to interior.

Line 585–589 I think this is an important topic, and I am really glad the authors are bringing it up. I hope they discuss the implications more, given paper such as Farinotti 2020, which discuss the growth of hydropower reservoirs following glacier retreat. This also has implications for GlOFs in other parts of the world.

We agree this is an important and interesting observation and now note the link to Farinotti et al. (2019) "Large hydropower and water-storage potential in future glacier-free basins".

Section 5.3 This section seems a bit problematic to me. Only a limited number of climatic variables were examined and the relationship between climate and glaciology is very non-linear (degree day model in the most basic sense). Does the winter precip account for more winter precip falling as rain? This is discussed in the later part of the section, but leads me to wonder why the issue was brought up in the first part of section. I suppose one motivation may be to discuss the role of topography, as opposed to climatology or glacier dynamics. However, lumping these three categories together presents something of a "chicken or the egg" problem. I recommend reconsidering this section. I discuss these issues in the cover letter.

We changed the title of this section to "lack of evidence for … climatic control" rather than "lack of climatic control" as it read before. It is correct that one could run correlations with other climatic variables, and we acknowledge the potential role of unexplored variables in Sec 5.4 . However, we do run a fairly large number of correlations and find minimal evidence for direct climatic control on lake area change, and believe the new framing is better supported by our data.

It is correct that we present this section partly to oppose the stronger findings for topographic controls. But we believe it is still valuable to present a result of no finding, such as we summarize

in this section, because links to climate are an obvious hypothesis to explain observed lake behavior. Therefore, this finding of minimal evidence for a link between climate and lake behavior is interesting in its own right. Not publishing negative results has been singled out as a major factor in the scientific "replication crisis" (e.g.,  https://www.nature.com/news/1-500-scientists-lift-the-lid-on-reproducibility-1.19970), and we view it to be especially important to publish a result of "no finding" in this context.

Additionally, we added a clarification in Sec 2.3 that the precipitation dataset does not distinguish between rain and snow. Therefore, this dataset would not be able to resolve a transition from dominantly snow to dominantly rain, which would likely negatively impact glacier mass balance, if the total winter precipitation were unchanged.

Line 605 backward climatic correlations... inverse? also the possibility for these relationships are discussed, but no confidence interval/or correlation statistic is given. This is problematic.

We changed "backwards" to "unintuitive". We just mean correlations that lack a clear physical mechanism underlying the association. The revised manuscript includes much more quantitative discussion of covariance between environmental variables, and this line now also points to figures and tables for support.

Line 615–626 Given the non-linear reaction of glacier dynamics to climate and the justification here, I am curious why climatic parameters were explored. It seems rather post-hoc to explain why climate matters little given the correlation is small. To me, this should have been accounted for when designing the experiment.

Climate and climate change are obvious first hypotheses for mechanisms to explain the observed lake area behavior. Admittedly, we were probably overly simplistic in our thinking of the link between climate, glaciers, and ice-marginal lakes when we originally set out on this study. However, it is very plausible that there are many researchers in the community who would expect that lakes have grown fastest in the places that have warmed most rapidly, which has caused the most dramatic rates of glacier mass loss. This study can show that this is not the case and that the story is more complicated. We view that as being a statement worth making, to remind others that there are a lot of moving pieces and nonlinear dynamics in these systems.

Lines 643 Be careful about GLOFs. These can also occur on moraine dammed lakes and while it is beyond my expertise, these dynamics could well evolve with changing proglacial lakes.

We revised this statement to include discussion of both ice-dammed and proglacial lakes as potential GLOF generators.

Section 5.5 I think these section may need restructuring. I believe much of its content is in some way discussed above or deals with the inherent limitations or advantages of physical vs statistical modeling.

Thank you for highlighting this issue. This section has been substantially revised to reduce redundancy with other sections (e.g., elaborating on "unintuitive correlations") and now instead devotes more discussion to the inseparability of associations with lake elevation, continentality, and initial area. The revised section more clearly conveys our difficulty in establishing causality between these potential controls on ice-marginal lake area change, as you have encouraged us to do in other comments.

Lines 663 Doesn't this sentence run counter to many of the arguments presented in lines 615–626?

We do not see these sections as being contradictory. Glaciers are certainly sensitive to climate (Line 663 in the original submission), though this relationship can be complicated, and the exact sensitivity/response can vary from glacier to glacier (Lines 615-626).

Section 6 I often consider "Conclusions" the best opportunity to position the research in the existing knowledge and state the knowledge gaps that have been filled. As a result, I am skeptical of the fact that no citations or references exist in this section.

This is a matter of opinion and writing style, and we choose to keep the conclusion as is.

Figures
Figure 1 I noticed this on lots of the figures, here especially. The lake area change is the close to the color of the glaciers. Can different colors be selected? also it seems a bit curious to me why lake area change, a result, is being presented here. I understand the desire to save space, but would another metric (lake area?) be better? "Detached lake" ... this seems like another term that should be defined together with the rest.

We changed figure colors to promote readability. Additionally, we now present two map-view figures so the revised Figure 1 just shows the location of proglacial and ice-dammed lakes, as opposed to showing the result of area change (now the separate Fig 3). Detached lakes are defined in Sec 3.1.

Figure 2 Please consider the colors again. Also, I understand the appeal of including this information. However, I am not entirely sure that I took away important findings from the figures and trends were hard to visualize given the layout. The authors could omit the figure if they desire.

We moved this figure to the supplements, as we agree it is not entirely necessary for the main text.

Figure 3 I like this figure. It demonstrates the important things. Would it be worth making a couple more panels (or a cartoon) with each type of lake? Proglacial, ice-dammed, detached...

Thank you. We did not add a schematic to this figure to avoid more complication to an already busy figure, but now include examples of both proglacial and ice-dammed lakes in Figs 1-2, which captures the essence of what you're suggesting.

Figure 4 Given the choice of having the three categories of lakes above (Section 3.1, I think). Would it make sense to add a third regression line with all lakes? May the divergent behavior of the two types of lakes here suggest that the "Ice-marginal" type of lake be omitted from analysis?

This is a good suggestion, but for the reasons listed above in the comment about Line 250, we believe it makes sense to not include an "all ice-marginal lakes" fit line. In addition, this plot is already somewhat busy as is, and the addition of more lines may make it confusing to many readers.

Figure 7 I would recommend presenting this information in 2 plots. One with percip/area change and one with temp./area change. To me this plot describes more the change in climate as opposed to the effect on lakes, which is hard to see amongst the different colors and shapes.

We moved this figure to the supplements, because it was somewhat distracting from the main points of this paper. As mentioned in the next comment, this paper is more about the lakes than how climate is changing, and so we have tried to minimize this kind of content in the main text.

Figure 8 Again, this is a somewhat difficult figure to read, and in my opinion somewhat deviates from the point of the paper, which is about lakes, not necessarily climate. Discerning a trend from the color bars is quite difficult for me, and the other information is quite intuitive and presented Figure 2. If the authors decide that this figure must stay, I recommend changing the c-axis and y-axis for the panels.

While we moved the other climate-related figures to the supplements (Figs. S1 & S5 in the revised manuscript), we do believe it is important to leave this figure in the main text. In the revised text, the figure appears later (now Fig. 9), after more text has been devoted to analyzing lake area change, emphasizing that these are our primary results. However, given how much we discuss "unintuitive links" between climate variables and lake area change, we believe this figure provides a more intuitive visual for putting forth physical mechanisms to explain these associations than just a table or test statistic value could provide. In the revised manuscript, we discuss this covariance between environmental variables in greater detail and believe this figure is now more relevant and better supported. While the new correlation matrix (Fig. 10) shows these values in a

more abstract way, we believe it may help get our point across to show the data in this more physical manner. We choose those variables for the y-axes because they are the ones most strongly linked to distance from the coast (the x-axis). The variables in the color axis generally do not monotonically vary with distance from the coast (aside from lake area change) and so the plot ends up not being as meaningful.

Figure 9 Proglacial new this seems like new term, possibly mentioned before. I think I understand what it is, but I recommend creating some kind of glossary early in the paper to make these things clear. Also the exclusion of "Ice-marginal" makes me think that there are two categories of lake, not                                                                                                                                                3.

We have added definitions of  this term (and reminders of its meaning) in Sec 3.2 ("lake area change analysis"), Sec 4.1 ("summary of regional lake area change") and the caption of Fig 3 (where the term first appears in a figure). These "new lakes" are lakes we observed to form during our study period. While we show them differently on plots, which allows the reader to see the environmental variables associated with new lake formation, we do not separate them for any statistical analyses.

Figure 9–11 Isn't this information a visual representation of the findings in Table 2? I think for the point of brevity and such that one or the other should be included. The other could fit well in a supplement.

It is correct that the essence of these figures is contained in Table 2. Table 2 includes all correlation results, while these figures just show a few selections (3/15 environmental variables). We believe that it is useful to show a few examples of the scatter plots because these can be more physically meaningful and/or intuitive than simply reporting a correlation coefficient. A reader might not be able to mentally picture what a $\tau = 0.3$ looks like, and showing these plots give a concrete example of how much predictive power exists in this relationship. We therefore choose to keep all these figures.